# Structural insights into sequence-dependent Holliday junction resolution by the chloroplast resolvase MOC1

Junjie Yan[1,2], Sixing Hong[1,2], Zeyuan Guan [1], Wenjing He[1], Delin Zhang[1] & Ping Yin[1✉]

Holliday junctions (HJs) are key DNA intermediates in genetic recombination and are eliminated by nuclease, termed resolvase, to ensure genome stability. HJ resolvases have been identified across all kingdoms of life, members of which exhibit sequence-dependent HJ resolution. However, the molecular basis of sequence selectivity remains largely unknown. Here, we present the chloroplast resolvase MOC1, which cleaves HJ in a cytosine-dependent manner. We determine the crystal structure of MOC1 with and without HJs. MOC1 exhibits an RNase H fold, belonging to the retroviral integrase family. MOC1 functions as a dimer, and the HJ is embedded into the basic cleft of the dimeric enzyme. We characterize a base recognition loop (BR loop) that protrudes into and opens the junction. Residues from the BR loop intercalate into the bases, disrupt the C-G base pairing at the crossover and recognize the cytosine, providing the molecular basis for sequence-dependent HJ resolution by a resolvase.

[1] National Key Laboratory of Crop Genetic Improvement and National Centre of Plant Gene Research, Huazhong Agricultural University, 430070 Wuhan, China. [2] These authors contributed equally: Junjie Yan, Sixing Hong. ✉email: yinping@mail.hzau.edu.cn

Holliday junctions (HJs) are a central intermediate formed during the process of homologous recombination[1], playing a critical role in promoting genetic diversity and repairing double-stranded DNA breaks[2]. HJs exhibit a cross-stranded four-way junction structure involving homologous pairing and DNA strand exchange[3,4], and must be timely eliminated to maintain faithful chromosome segregation and genome stability[5]. Defective HJ resolution would lead to aberrant chromosome morphology, resulting in tumorigenesis in humans[6,7].

One of the key pathways for HJ resolution is carried out by a family of structure-selective endonucleases called HJ resolvases, which have been identified from a wide variety of organisms including bacteriophages, bacteria, archaea, yeasts, plants, and mammals[8–12]. HJ resolvases have been extensively studied in lower organisms including phage T4 and T7 endonucleases[13,14], bacterial RuvC[15], and archaeal Hjc[16]. In eukaryotes, the nucleus-encoded Gen1 and its orthologs have been broadly investigated[6,11,17–20]. In addition, HJ resolvases have also been identified in mitochondria and chloroplasts, including Cce1[21], Ydc2[22], and MOC1[23].

These resolvases exhibit distinct cleavage characteristics. T4 and T7 endonucleases display no sequence specificity for cleavage. In contrast, sequence-dependent HJ resolution was reported for RuvC (5′-A/TTT↓G/C-3′)[8,15,24], Cce1 (5′-ACT↓A-3′)[25,26], Ydc2 (5′-C/TT↓−3′)[27,28], and MOC1 (5′-C↓C-3′)[23]. Crystal structures of HJ-bound T4 endo VII and T7 endo I revealed that the enzymes interact exclusively with the backbone of DNA and form no hydrogen bonds with the DNA bases[13,14], which explains their lack of sequence specificity. The complex structure of TtRuvC-HJ was obtained in a noncatalytic state and no direct base-recognition was observed[29], which does not explain the molecular basis of the cleavage specificity. To date, the molecular basis of sequence-dependent HJ resolution by resolvase remains largely unknown.

In this study, we determine the crystal structures of HJ-free *Zea mays MOC1* (ZmMOC1), *Nicotiana tabacum* MOC1 (NtMOC1), and HJ-bound NtMOC1 at resolutions of 2.5 Å, 2.0 Å, and 2.5 Å, respectively. MOC1 exhibits dimerization and an RNaseH fold, belonging to the retroviral integrase superfamily. The HJ is embedded into the basic cleft of the dimeric enzyme. Four acidic residues constitute the catalytic tetrad that acts on the cleavable phosphodiester bond. Moreover, we characterize a base recognition loop (BR loop) that protrudes into the junction and disrupts the C–G base pairs at the crossover. Residue D183 from the BR loop recognizes the cytosine, determining the cytosine-dependent HJ resolution.

## Results

### Chloroplast MOC1s exhibit cytosine-dependent HJ resolution.

Recently, mutational screening analysis identified the chloroplast-localized HJ resolvase MOC1 that is essential for chloroplast nucleoid segregation in *Chlamydomonas reinhardtii* and *Arabidopsis thaliana*[23]. Dysfunction of MOC1 resulted in aberrant nucleoid morphology, indicating a crucial role for this kind of gene in plant chloroplast development. At the biochemical level, AtMOC1 introduced a cleavage between two consecutive cytosine (5′-C↓C-3′) residues at the core of the HJs[23]. MOC1 homologs are widely present across the plant kingdom with a highly conserved C-terminal domain (Supplementary Fig. 1). In this study, we purified homologous MOC1s from various plant species including *A. thaliana* (AtMOC1), *Glycine max* (GmMOC1), *Gossypium raimondii* (GrMOC1), *N. tabacum* (NtMOC1), *Oryza sativa* (OsMOC1), and *Z. mays* (ZmMOC1).

To examine the HJ cleavage specificity of these MOC1 orthologs, a synthetic bimobile HJ of a 2-base-pair (2-bp) homologous core

with a CCGG core (Supplementary Fig. 2), termed X2 (CCGG), was used as a control according to a previous study[23]. In addition, six additional X2 variants with mutations in the homologous cores were used as the substrates (Fig. 1a). All these MOC1s exhibited obvious HJ resolution activity toward the X2 (CCGG) substrate (Fig. 1b), similar to a previous investigation[23], indicating cleavage between the consecutive cytosine residues (5′-C↓C-3′). In addition, weak cleavage activity against X2 (CGCG) and X2 (CATG) were also observed for some orthologous MOC1s (Fig. 1b). Further cleavage site mapping experiments using AtMOC1 revealed symmetrical cuttings at the crossover in the sequence 5′-C↓G-3′ for the X2 (CGCG) substrate and 5′-C↓A-3′ for the X2 (CATG) substrate (Supplementary Fig. 3). These results revealed that MOC1s resolve HJs in a cytosine-dependent manner (after a cytosine).

### Crystal structure of MOC1.

To elucidate the molecular mechanism of cytosine-dependent HJ resolution by MOC1s, we then performed a systematic crystal screening for these homologous MOC1s. We first determined the crystal structure of HJ-free ZmMOC1 (T107-V280) via iodide-based single-wavelength anomalous diffraction (I-SAD) at a refined resolution of 2.5 Å (Methods and Supplementary Fig. 4a-b) and the structure of a quadruple mutant NtMOC1 (N108-S275, I112V/Q162K/E235Q/239Q) at a 2.0 Å resolution (Fig. 2a, Supplementary Table 1). Structural alignment of NtMOC1 with ZmMOC1 revealed a similar fold with an RMSD value of 0.919 Å (Supplementary Fig. 4c). Moreover, the crystal structures of the HJ-bound NtMOC1 were also determined. Thus, we use NtMOC1 as a representative to demonstrate the structural features of MOC1 in the following section.

NtMOC1 is a homodimer (Fig. 2b), akin to the known resolvases[18,19,30–38]. For each protomer, the core of NtMOC1 is formed by five-stranded β-sheets (β1β2β3β6β7) composed of three antiparallel (β1β2β3) and two parallel strands (β6β7)—a characteristic feature of the RNase H fold[39], belonging to the retroviral integrase superfamily[39]. The β-sheets are flanked by six α helices (α1 and α2 on one side, and α3, α4, α5 and α6 on the other side) (Fig. 2b, c). The α-helix 2 (α2) from each protomer constitutes a helical bundle, where G192, G196, G200, and A204 form a hydrophobic patch on the dimer interface (Fig. 2d, e; Supplementary Fig. 5a). Both analytical ultracentrifugation (AUC) and size-exclusion chromatography (SEC) experiments revealed that double mutations of G200E and A204E could disrupt dimer formation (Supplementary Fig. 5b, c). The SEC results further indicate that single mutation of G200E or A204E might partially disrupt the dimerization (Supplementary Fig. 5c). Although these mutants retain some HJ-binding ability, their HJ cleavage activity is drastically impaired (Supplementary Fig. 5d, e), demonstrating that NtMOC1 functions as a dimer.

Both protomers of apo NtMOC1 revealed identical architectures with no structural variation (Supplementary Fig. 6a). A structural homolog search by Dali[40] revealed that NtMOC1 has the highest structural similarity with RuvC resolvase (Supplementary Table 2). In addition, NtMOC1 displays some similarity to Cas9, due to the existence of a RuvC nuclease domain in the protein[41]. Structural superimposition of NtMOC1 with RuvC revealed similar protomers (with an RMSD value of 3 Å) and dimeric architectures (Supplementary Fig. 6b, c), suggesting their potentially similar HJ binding and resolution activities.

### Crystal structure of NtMOC1 in complex with HJ.

To determine the complex structure, a series of HJs with varying arm lengths and homologous cores were used for cocrystallization

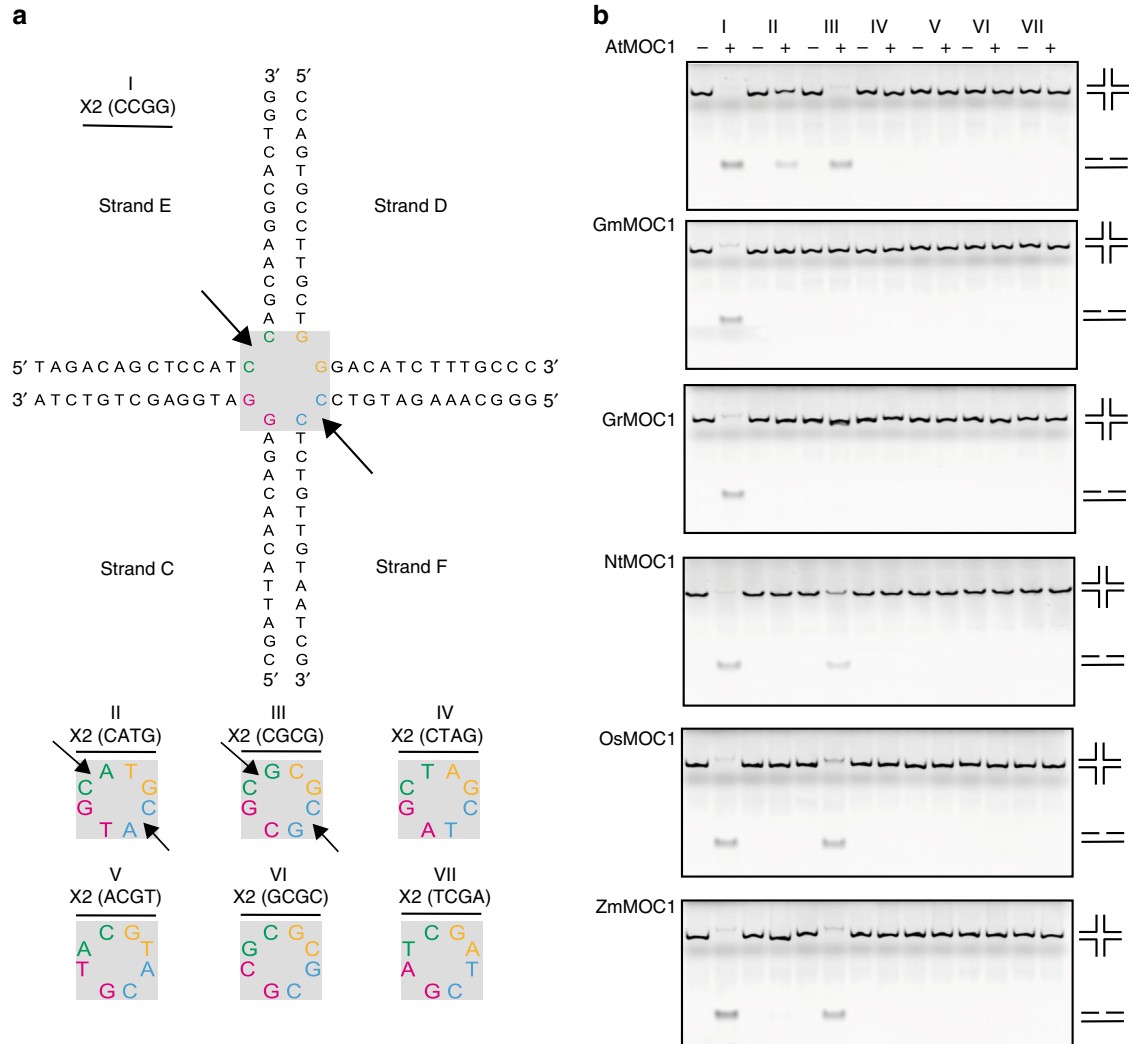

**Fig. 1 Sequence-dependent HJ cleavage by chloroplast MOC1s. a** Schematic drawing of the HJ substrate X2. X2 was prepared by annealing 4 DNA oligos. The upper panel shows the structure of X2 (CCGG). Two homologous base pairs are shaded in gray. The lower panel indicates X2s with variant CCGG cores. These X2s share identical nucleotides as X2 (CCGG) except the 2 bp homologous cores. These seven X2s are numbered sequentially (I–VII). Arrows indicate the cleavage site. **b** The sequence-specific HJ resolution activity of MOC1 orthologs was examined. AtMOC1, GmMOC1, GrMOC1, NtMOC1, OsMOC1, and ZmMOC1 are MOC1s from *Arabidopsis thaliana*, *Glycine max*, *Gossypium raimondii*, *Nicotiana tabacum*, *Oryza sativa*, and *Zea mays*, respectively. The final HJ concentration in each lane is 250 nM. The final protein concentration of MOC1 is 1000 nM. The reactions were examined by native PAGE and visualized by GelRed staining. Labels on the top indicate the different X2 substrates used as shown in **a**. Source data are provided as a Source Data file.

with MOC1s. Most crystals were either poorly diffracted or showed no DNA density after data processing. After tedious trials, we finally determined two crystal structures of the NtMOC1-HJ complex at an approximately 2.5 Å resolution in the P2₁2₁2 space group (Supplementary Table 1). One of the structures contains the active NtMOC1 and HJ with a noncognate CATG core sequence, and the other is composed of a cleavage-inactive NtMOC1 mutant (D116A/E175A/D253A/E258A) and an HJ with a cognate CCGG core sequence (Supplementary Fig. 7). The HJs in both structures are four-stranded with a 9-bp arm length in each direction with only variation in the crossover with either the CATG core or the CCGG core (Supplementary Fig. 8a, b). These two NtMOC1-HJ complexes exhibit a nearly identical overall fold (Supplementary Fig. 8c). The present and previous[23] cleavage site mapping experiments revealed the phosphodiester bond within the CC or CA bases at the crossover is cleavable (Supplementary Fig. 2). In the HJs of both complexes, the CATG and CCGG at the branch point were well aligned (Supplementary Fig. 8d).

Here, we use the complex structure of NtMOC1-HJ with a CATG core sequence as a representative to demonstrate the catalytic and binding features. NtMOC1 in the complex also exhibits dimerization, and the HJ is embedded into the cleft of the dimeric NtMOC1 (Fig. 3a). Structural alignment of the apo NtMOC1 with HJ-bound NtMOC1 revealed minor conformational changes with an RMSD value of 0.527 Å over 242 Cα (Supplementary Fig. 9), which mainly arose from the loop preceding α2 (we termed this loop the base recognition loop (BR loop), discussed below). An additional pair of unique antiparallel β-sheets (β4β5) was observed only in the complex structure flanking the dimer interface but the density corresponding to these residues in the apo structure is missing (Fig. 2b and Fig. 3a), suggesting a role for this segment in DNA interactions.

**Catalytic center of NtMOC1.** In the HJ-bound complex structure, HJ exhibits an open planar and X-shaped conformation with an overall two-fold symmetry (Fig. 3b). The phosphate backbones of the non-exchanging strands are antiparallel, whereas the

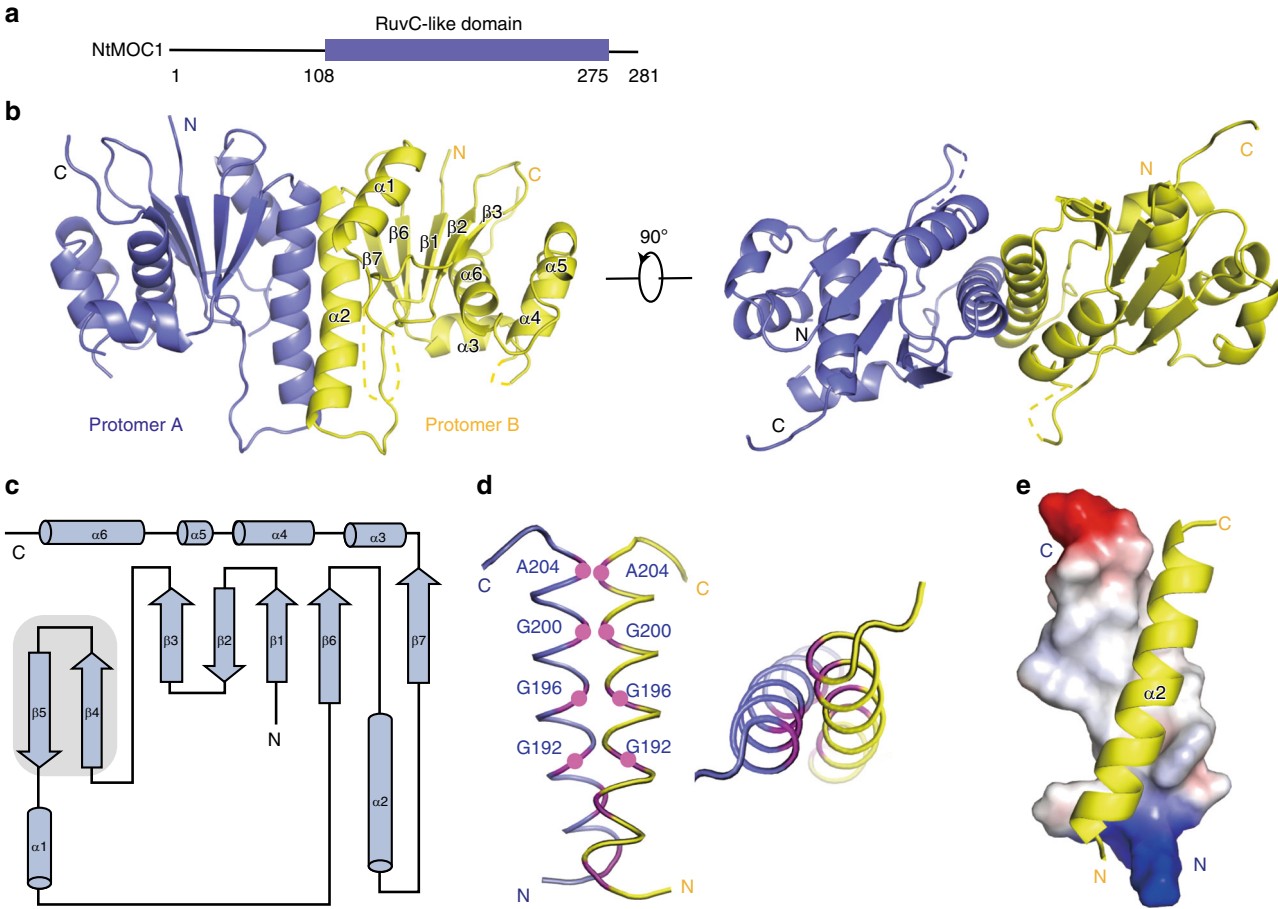

**Fig. 2 Crystal structure of apo NtMOC1. a** Schematic domain structure of NtMOC1. The C-terminus (residues 108-275) is the RuvC-like domain, the structure of which was determined in this study. **b** Two perpendicular views of the overall structure of apo NtMOC1. Secondary structural elements are labeled in protomer B. Dashed lines indicate residues with missing electron density. **c** Topological diagram of the NtMOC1 protomer. Secondary structure elements are labeled as in **a**. Two β-sheets (β4–β5) observed in the complex structure are shaded in gray. The electron density of this segment is missing in the apo structure. **d** Residues with small side chains constitute the dimer interface. **e** Dimer interface of the helix bundle. α2s from protomers A and B are shown in the electrostatic surface potential and in the yellow cartoon illustration, respectively. The blue and red colors indicate positively and negatively charged residues, and white regions indicate hydrophobic residues.

phosphate backbones of the exchanging strands are sharply kinked at the crossover to form a U-turn. The center of the junction is opened by NtMOC1 into a parallelogram with cross dimensions of 32 Å between the non-exchanging strands and 9 Å between the exchanging strands (Fig. 3b). The scissile phosphates are in close proximity to the two active sites of the dimeric enzyme (Supplementary Fig. 10a). Four acidic amino acids, namely D116, D118, E175, and E258, coordinate a magnesium ion approach to the cleavable phosphodiester backbone, constituting the catalytic tetrad (Fig. 3c). Alanine substitutions of D116, D118, E175 and E258 completely abolished HJ cleavage activity, although the HJ binding activity was retained (Fig. 3d, e). As a control, mutation of residue D253 neighboring the active site showed no effect on either HJ binding or HJ cleavage activity compared with the wild-type NtMOC1 (Supplementary Fig. 10b–d).

**Protein-DNA contacts**. The complex structure reveals extensive interactions between dimeric NtMOC1 and four DNA strands with a global contact area of 2595 Å² (Fig. 4a, Supplementary Fig. 11). To clearly demonstrate the protein-DNA interactions, we defines the four DNA strands (C, D, E and F). Each strand harbors 18 nucleotides numbered sequentially from the 5′ side. The arm length in each direction is 9 bp (Supplementary

Fig. 8a–b), and the phosphodiester to be cleaved is located at the crossover between cytosine 9 (C9) and adenine 10 (A10) at the continuous strands E and F based on the cleavage site mapping[23] (Supplementary Fig. 2), complex structure alignment (Supplementary Fig. 8), and its proximity to the catalytic tetrad (Fig. 3c).

Electrostatic potential analysis revealed that a high proportion of positively charged amino acids are distributed on the surface toward the HJ (Fig. 4a). The residue R149 in β5 makes direct contacts with the neighboring DNA strands (Fig. 4b). The amine group of R149 (protomer B) interacts with the phosphate group of nucleotides G12 (chain E) and A12 (chain D) (Fig. 4b). The K185 residues from both protomers are located at the N-terminus of the dimeric helix bundle, protruding into the junction, interacting with the phosphate group of the flipped-out nucleotide G10 from the non-exchanging strands (Fig. 4c). K218 (protomer A) from α5 contacts the phosphate group of nucleotide C9 (chain F), and the neighbored K225 (Protomer A) contacts the phosphate group of nucleotide C8 (chain F) via its main chain amine group (Fig. 4d). Three consecutive basic residues (R250/K251/K252) at the periphery of the dimeric enzyme contact the exchanging DNA strands (Fig. 4e). These interactions were verified by mutational analysis. Charge reversal mutants with single mutations of these residues (R149D, K185D, and K218D) retained equivalent HJ binding activity (Supplementary Fig. 12a). However, the HJ

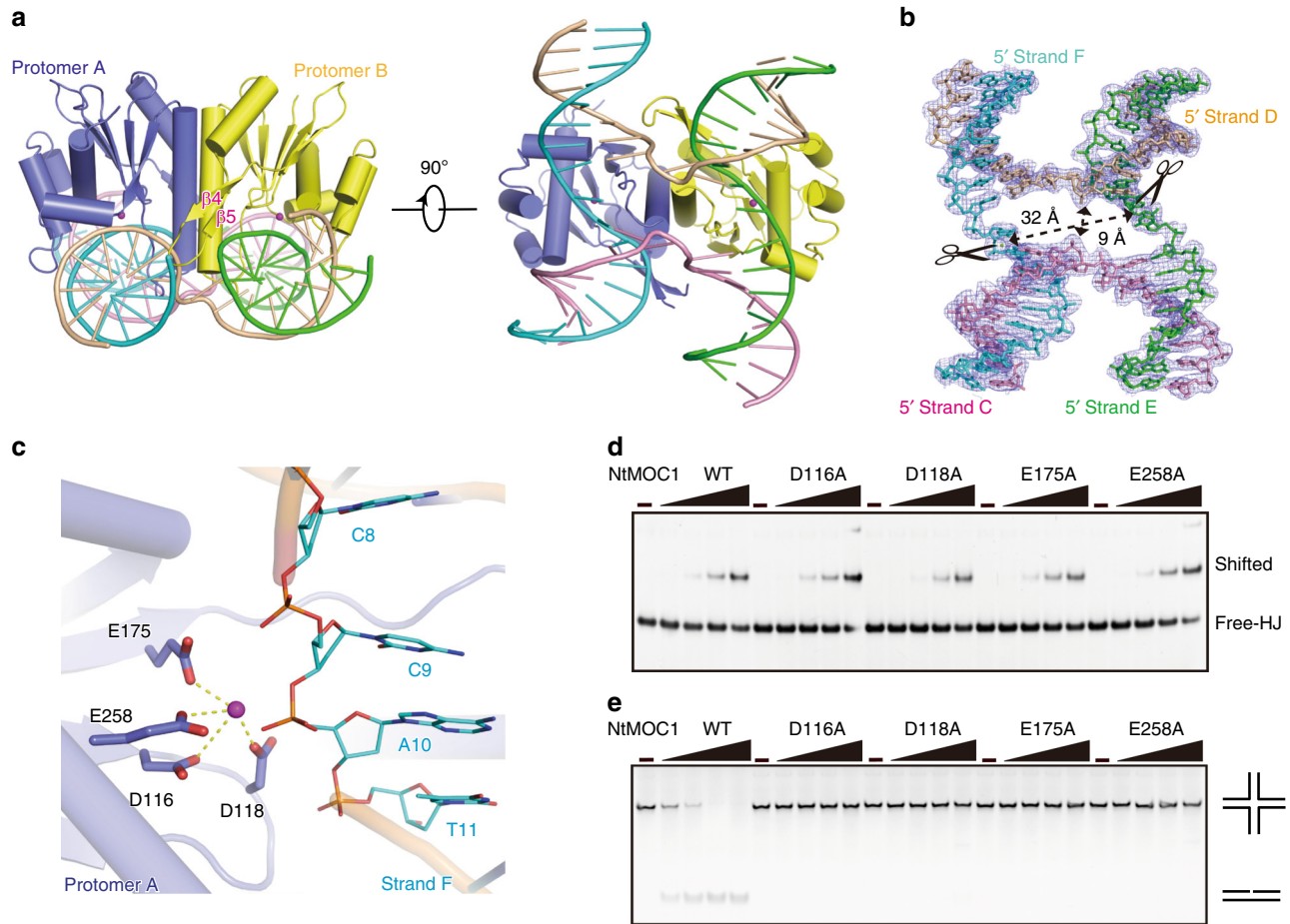

**Fig. 3 Crystal structure of NtMOC1 in complex with a synthetic HJ. a** Two perpendicular views of the complex structure. Subunits of the protein dimer are shown in slate and yellow. The four DNA strands C, D, E, and F are colored in pink, wheat, cyan, and green, respectively. The HJ is embedded into the cleft of the dimeric enzyme. **b** Structure of the HJ in the complex. The electron density map is contoured at 1.0 σ. Scissors indicate the scissile phosphates. Distances between the exchanging and non-exchanging strands are measured. **c** Active centers of NtMOC1. The magnesium ions are shown as purple spheres. Four acidic amino acids, coordinating with magnesium, constitute the catalytic tetrad. **d** Mutations at the catalytic tetrad of NtMOC1 have no effect on the HJ binding activity. **e** Mutations at the catalytic tetrad of NtMOC1 abolish the HJ cleavage activity. For the HJ binding and cleavage assay, X2 (CCGG) was used as HJ substrate. The final concentration of HJ in each lane was 250 nM. Five gradients with increasing concentrations (0, 125, 250, 500, and 1000 nM) were applied for each protein sample. The reactions were resolved by native PAGE and visualized by GelRed staining. Source data are provided as a Source Data file.

cleavage activity of R149D, K185D and K218D was completely lost (Supplementary Fig. 12b), suggesting that these three residues are essential for maintenances of the HJ in a favorable configuration for cleavage. The control mutant K225D retains the HJ cleavage activity (Supplementary Fig. 12b). Simultaneous mutation of R149D/K185D/K218D/K225D (Tetra-M) or R250/K251/K252D could result in total loss of HJ binding and cleavage activity (Supplementary Fig. 12).

**Molecular basis of cytosine-dependent HJ resolution by MOC1.** Close inspection of the HJ-bound structure revealed how MOC1 cleaved HJ in a cytosine-dependent manner (5′-C↓C/G-3′). Two amino acids (Y180 and D183) in the loop preceding α2 are involved in sequence-specific HJ resolution (Supplementary Fig. 13). Hereafter, we refer to this loop as the base recognition loop (BR loop). Residue Y180 intercalates into the bases one nucleotide prior to the cleavage site (Fig. 5a), forming stacking interactions with the flanking nucleotide bases, similar to the aromatic residue F69 observed in the corresponding loop of *E.coli* RuvC (F73 in *T. thermophilus*)[29,42,43]. More importantly, D183 specifically recognizes the cytosine (C9) via a hydrogen bond (Fig. 5a). Alanine substitution of D183 (D183A) results in

complete abolishment of HJ cleavage activity, although the HJ binding activity is only partially decreased (Fig. 5b, c). Mutation of Y180 (Y180A) has little effect on HJ binding and resolution activity (Fig. 5b, c). Although the overall structures of apo and HJ-bound NtMOC1 exhibited similar architectures (Supplementary Fig. 9), residues Y180 and D183 in the BR loop underwent striking conformational changes, demonstrating the great flexibility of the BR loop (Fig. 5d).

The pairing of D183 (protomer A) to the cytosine (C9, from strand F) led to flipping out of the initially paired guanine (G10, from strand D) from the duplex (Fig. 5a, Supplementary Fig. 12). The flipped-out guanine (G10) is further stabilized by hydrogen bond networks with amino acids Q186 and G187 of helix 2 (α2) from protomer B and the surrounding nucleotides (Fig. 5a). This MOC1-induced guanine flipping is reminiscent of other base flipping phenomena including epigenetic regulation of DNA and base excision repair by some modification-related nucleases, such as HhaI methyltransferase[44] and uracil-DNA glycosylase[45].

Based on the structural and biochemical results, we propose a working model of MOC1 (Fig. 6). In general, HJ forms a compact structure with all base pairings[3,4], but is distorted when the dimeric MOC1 penetrates into the junction and disrupts the C–G

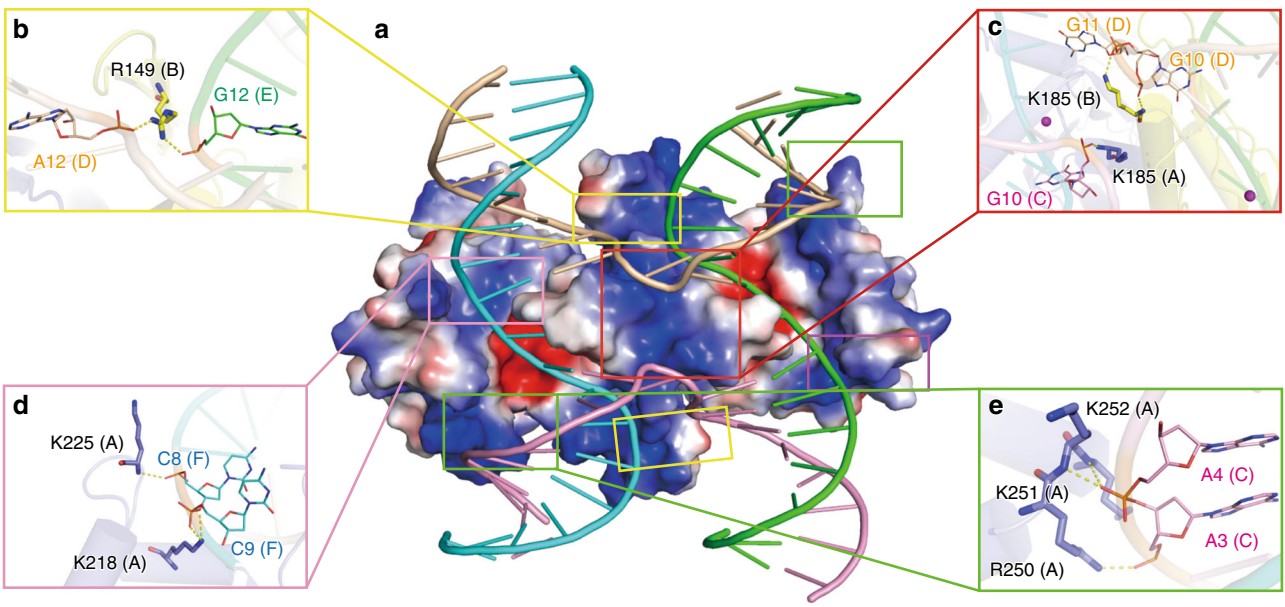

**Fig. 4 Protein-DNA contacts. a** The bottom view of the complex shows the interaction surface. The protein dimer is shown by the electrostatic surface potential. Four DNA strands of the HJ are shown with cartoon illustrations. Four distinct interaction regions are labeled with colored rectangles. **b–e** Enlarged image showing the details of the interaction between NtMOC1 and the DNA strands. Capital letters in parentheses indicate corresponding protein or nucleotide chains. A and B represent protomer A and B; C, D, E, and F represent DNA strands C, D, E and F, respectively.

base pairing at the crossover prior to the cleavage-site. To fulfill sequence-dependent cleavage, D183 in the BR loop specifically recognizes the cytosine via a hydrogen bond, favoring hydrolysis of the phosphodiester bond by the catalytic tetrad in a cytosine-dependent manner.

## Discussion

HJs are important DNA intermediates in DNA damage repair and genetic recombination processes, and must be timely resolved into duplex DNA to maintain genome stability. Several kinds of resolvases possess the ability to resolve the HJ in a sequence-specific manner. HJs are highly dynamic structures that could undergo conformer exchange and branch migration, the relationship between HJ dynamics and sequence-specific resolution is largely unknown. Using single-molecule fluorescence resonance energy transfer (smFRET), a recent study indicated that HJs remain dynamic in the presence of resolvases (as exemplified by T7 endo I, RuvC, GEN1, and hMus81-Eme1)[46]. Due to the multivalent interactions between the dimeric enzymes and HJs, the resolvase-HJ complex could go through a partially dissociated intermediate state whereby the dynamic process of conformer exchange and branch migration can proceed without full dissociation[46]. In this study, we present the crystal structures of apo and HJ-bound chloroplast resolvase MOC1 and identify a crucial BR loop in MOC1 that contributes to the sequence-dependent HJ resolution. We speculate that the presence of MOC1 might not restrain the dynamics of the HJ, which can undergo branch migrations, extending or shortening the length of DNA hetero-duplex. Once the cognate sequence of CCGG (or CGCG) moves to the branch point, residues Y180 and D183 from the dynamic BR loop of NtMOC1 could penetrate into the junction and disrupt the C–G base pairs. Meanwhile, the disrupted cytosine is recognized by D183, generating a favorable HJ configuration, thus facilitating the hydrolysis of the phosphodiester bond by the catalytic tetrad of NtMOC1 (Fig. 6 and Supplementary Fig. 14a).

Previous studies indicated that RuvC (5′-A/TTT↓G/C-3′)[8,15,24], Ydc2 (5′-C/TT↓−3′)[27,28], and Cce1 (5′-ACT↓A-3′)[25,26] are all sequence-specific resolvases that introduce a nick after a thymine.

Crystal structures of RuvC[29,30,43] and Ydc2[33] revealed their RNase H fold, belonging to the retroviral integrase family. Furthermore, these proteins function as dimers, harboring a cleft in each protomer that accommodates binding duplex DNA. Biochemical studies suggested that the presence of RuvC or Ydc2 should disrupt the base pairing of HJ at the crossover[15,47,48], as evidenced by the MOC1 structure. A similar long loop is present preceding the α-helix that forms a helical bundle in RuvC and Ydc2 (Supplementary Fig. 14b). Moreover, the loop contains a consensus sequence: $E(X)_{2-4}Y/F(X)_{1-2}D/K/R/N/Q/S$. The catalytic glutamate is located at the tip of the loop (Supplementary Fig. 14c). Within the loop, aromatic residues such as tyrosine or phenylalanine are generally present (Supplementary Fig. 14d). Following the aromatic residue (Tyr or Phe), there are charged (Asp, Lys, Arg) or polar (Asn, Gln, Ser) residues (Supplementary Fig. 14d), which might mediate specific base recognition. Thus, we speculated that these resolvases might adopt a similar mechanism by directly recognizing the thymine via certain residues to establish sequence-specific HJ resolution similar to the activity of MOC1.

During the submission and review of our manuscript, two related works regarding sequence-specific HJ resolution were reported. In one study, the complex structure of DNA-bound TtRuvC at 3.4 Å was reported[49]. Via molecular dynamics simulations, disruption of the scissile T-A base pairs at the crossover and the flipping out of adenine was observed[49]. Meanwhile, residue R76 was identified to form either stacking or H-bond interactions with the bases of the disrupted T-A base pair. This observation is consistent with our speculation regarding the crucial role of the BR loop in base recognition, as R76 is located in the BR loop (Supplementary Fig. 14d). Another study presented the crystal structures of apo and HJ-bound ZmMOC1[50]. The structure of ZmMOC1 in this study is nearly identical to the recently reported ZmMOC1 structure (PDB: 6IS9) with an RMSD of 0.45 Å (Supplementary Fig. 15a). Although the HJs were annealed using different sequences and strategies (4 oligos in our study and 2 oligos with a T-loop in the other study), superimposition of the complex structure of NtMOC1-HJ and

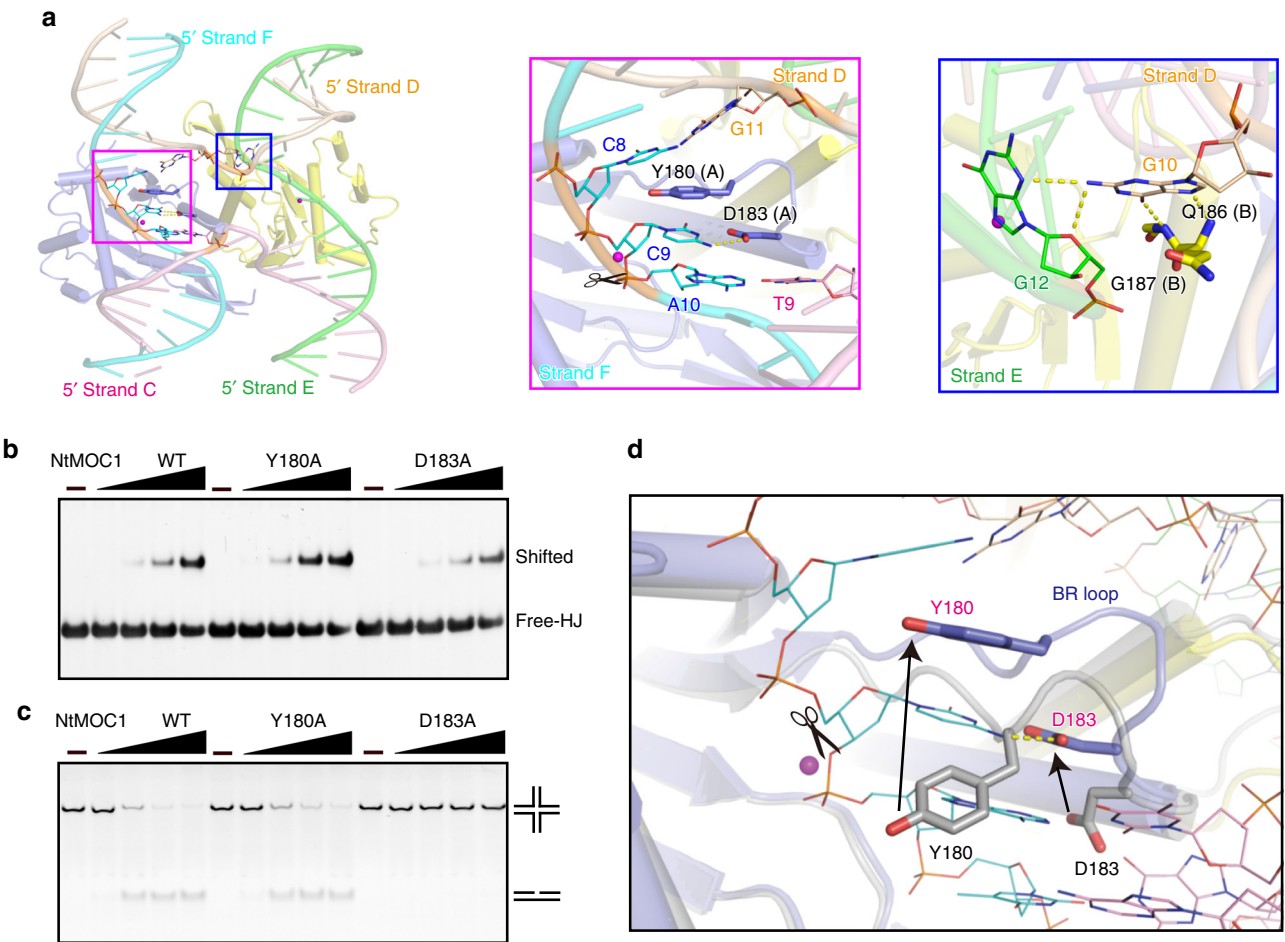

**Fig. 5 Molecular basis of cytosine-dependent HJ resolution by NtMOC1. a** The C–G base pairs at the crossover are disrupted by NtMOC1. Left panel: Overall view of the disrupted C–G base pairs in the complex. Middle panel: The disrupted cytosine (C9) interacts with D183 via a hydrogen bond. Right panel: The flipped-out guanine is stabilized by residues and nucleotides. **b, c** HJ binding (**b**) and cleavage (**c**) activity examined for NtMOC1 with alanine substitutions of Y180 and D183. For the HJ binding and cleavage assay, X2 (CCGG) was used as HJ substrate. The final concentration of HJ in each lane was 250 nM. Five gradients with increasing concentrations (0, 125, 250, 500, and 1000 nM) were applied for each protein sample. The reactions were resolved by native PAGE and visualized by GelRed staining. Source data are provided as a Source Data file. **d** The BR loop undergoes a striking conformational change. Apo NtMOC1 is shown in the gray cartoon illustration. The protomers of HJ-bound MOC1 are shown in slate and yellow cartoons, respectively. DNA strands are shown with lines. The purple circle indicates the magnesium ion in the complex structure. The yellow dashed line indicates the hydrogen bond formed between D183 and cytosine. Arrows indicate the displacement of Y180 and D183. Residues in the apo and complex structures are labeled in black and magenta, respectively. The scissors indicate the cleavage site.

ZmMOC1-HJ exhibited comparable folds with an RMSD of 0.762 Å, suggesting a similar DNA recognition mode (Supplementary Fig. 15b). The C–G base pairs at the crossover of these HJs are similarly disrupted (Supplementary Fig. 15c). Moreover, the critical residues Y180 and D183 in the BR loop (BR motif) of NtMOC1 and the corresponding residues F179 and D182 in ZmMOC1 could be perfectly superimposed (Supplementary Fig. 15d), suggesting a general molecular basis for sequence-specific HJ resolution by the MOC1 orthologs.

Collectively, our work on NtMOC1 and studies by other groups' on RuvC[49] and ZmMOC1[50] could serve as important foundations for understanding sequence-specific HJ resolution by other as-yet-identified resolvases.

## Methods

**Molecular cloning, protein expression, and purification.** Homologous MOC1s from *Arabidopsis thaliana* (AT2G26840.1), *Glycine max* (XP_003532877.2), *Gossypium raimondii* (XP_012444423.1), *Nicotiana tabacum* (XP_016490038.1), *Oryza sativa* (LOC_Os01g16340.1), and *Zea mays* (ONM31293.1) were cloned into a modified pET15D vector with a 6×His tag fused at the N terminus. The DNA mutants were constructed by overlapping PCR. All the constructs were verified by

DNA sequencing. The plasmid was transformed into *E. coli* BL21 (DE3). One litre lysogeny broth medium supplemented with 100 mg ml$^{-1}$ ampicillin was inoculated with a transformed bacterial pre-culture and shaken at 37 °C until the cell density reached an $OD_{600}$ of approximately 1.0−1.2, protein expression was induced with 0.2 mM isopropyl-β-D-thiogalactoside at 16 °C for 12-16 h. The cells were collected by centrifugation, homogenized in buffer A (25 mM Tris–HCl, pH 8.0, 150 mM NaCl), and lysed using a high pressure cell disrupter (JNBIO, China). Cell debris was removed by centrifugation at 20,000 × *g* for 1 h at 4 °C, and the supernatant was loaded onto a column equipped with Ni$^{2+}$ affinity resin (Ni-NTA, Qiagen), washed with buffer B (25 mM Tris–HCl, pH 8.0, 150 mM NaCl, 15 mM imidazole), and eluted with buffer C (25 mM Tris–HCl, pH 8.0, 500 mM NaCl, 250 mM imidazole). The 6×His tag was removed through DrICE digestion. The protein was two-fold diluted to reduce the salt concentration and then separated by HiTrap Heparin (GE Healthcare) using a linear NaCl gradient in buffer A. The purified protein was concentrated and subjected to gel filtration chromatography (Superdex-200 Increase 10/300, GE Healthcare) in a buffer containing 25 mM Tris-HCl, pH 8.0, 500 mM NaCl, and 5 mM dithiothreitol. Purity of the proteins was examined using SDS-PAGE and visualized by Coomassie blue staining through all purification processes. The peak fractions were collected and stored at −80 °C. The mutant proteins were purified similarly as the wild-type proteins.

**DNA sample preparation.** The sequences of the oligo-nucleotides used to form the HJ substrates were synthesized (TianYi HuiYuan). To make the X2 (CCGG) HJ, the following four DNA strands (5′-GGGCAAAGATGTCCCTCTGTTGT

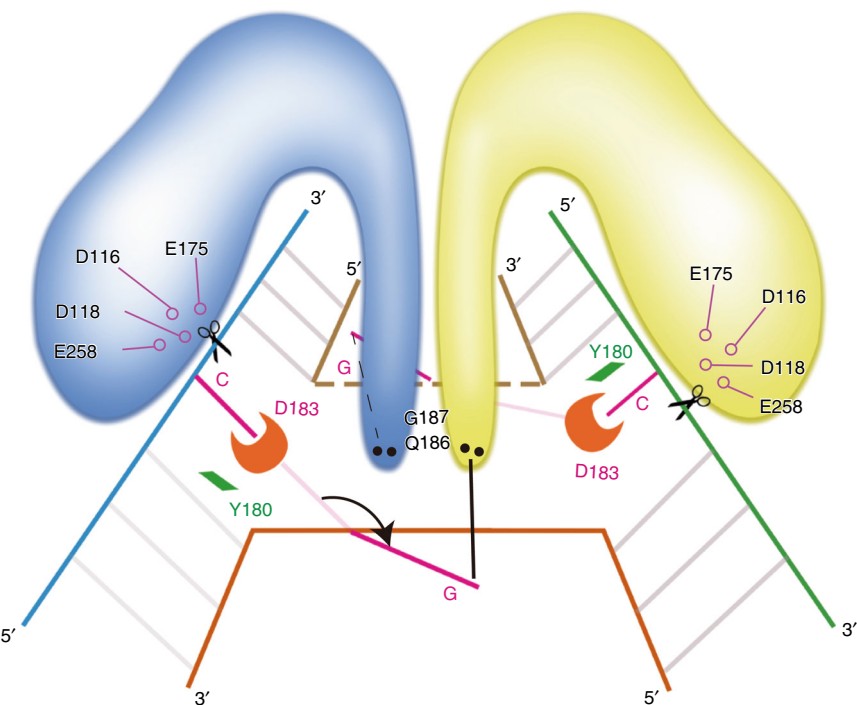

**Fig. 6 Proposed working model for cytosine-dependent HJ resolution by NtMOC1.** The blue and yellow models indicate the protein protomers. The dimeric enzyme penetrates into the junction, and the two residues Y180 and D183 from the BR loop protrude into the duplex and disrupt the base pairing at the crossover. Y180 intercalates into the bases, whereas D183 recognizes the cytosine prior to the cleavage site. The flipped-out guanine is further stabilized by residues from the N-terminal helical bundle (Q186 and G187). Four acidic residues constitute the catalytic tetrad, hydrolyzing the phosphodiester bond in a cytosine-dependent manner.

AATCG-3′; 5′-CCAGTGCCTTGCTGGGACATCTTTGCCC-3′; 5′-TAGACAG CTCCATCCAGCAAGGCACTGG-3′; 5′-CGATTACAACAGAGGATGGAGCTG TCTA-3′. The underlined nucleotides indicate the 2-bp homologous cores at the crossover) were resolved in lysis buffer (25 mM Tris-HCl, pH 8.0, 150 mM NaCl) and mixed at an equimolar ratio and incubated in boiled water and gradually cooled to room temperature. Several X2 (CCGG) variants were derived similarly by mutation of the CCGG core sequences. To obtain co-crystals, a series of truncated HJs with varying arm length and CCGG core sequences were similarly annealed. The individual oligo sequences used for HJ annealing are deposited in Supplementary Table 3.

**Crystallization.** Crystallizations were performed using the sitting-drop vapor diffusion method at 18 °C by mixing equal volumes (1 μl) of protein with the reservoir solution. For the apo MOC1, protein was concentrated to 10 mg ml$^{-1}$ and a final concentration of 1 mM MgCl$_2$ was added to the sample before crystallization trials. The rodlike crystals of ZmMOC1 (T107-V280) were grown in a well buffer containing 14% PEG3350, 0.15 M NH$_4$Cl and grew to full size within 3 days. High-quality ZmMOC1 crystals were soaked in the reservoir solution plus 0.1 M 5-amino-2,4,6-triiodoisophthalic acid (I3C, Molecular Dimensions) for 2 h. For NtMOC1, the tetra-mutant (N108-S275, I112V/Q162K/E235Q/E239Q) gave rise to crystals. The diamond shape crystal appeared overnight and grew to full size within three days in the well buffer containing 0.1 M Bis-Tris, pH 6.3, 22% PEG3350, 0.2 M Li$_2$SO$_4$.

To obtain MOC1-HJ complex, a final concentration of 1 mM MgCl$_2$ was added to the dimeric MOC1, which was further mixed with HJ DNA with a molar ratio of 1: 1.2. Numerous HJs with varying arm length and homologous cores were used to incubate with MOC1 to screen cocrystals. Most of the crystals of ZmMOC1-HJ complex were either poorly diffracted or showed no DNA density after data processing. After extensive and tedious trials, tetrahedron crystals from HJ with an arm length of 9 bp in each direction with CATG homologous core sequences in complex with NtMOC1 (N108-S275, I112V/Q162K/E235Q/E239Q) were appeared in a buffer containing 0.1 M Sodium acetate trihydrate, pH 5.0, 8.5% PEG3350, 2% Tacsimate, pH 4.0. And crystals of HJ with an arm length of 9-bp with CCGG homologous core sequence in complex with cleavage inactive NtMOC1 mutant (N108-S275, I112V/Q162K/E235Q/239Q/D116A/E175A/D253A/E258A) were appeared in a buffer containing 0.1 M sodium acetate trihydrate, pH 5.6, 15% 3-Methyl-1,5-Pentane Diol, 2% PEG4000, 2% D-sorbitol. Crystals were harvested and flash-frozen in liquid nitrogen and cryoprotected by adding glycerol to a final concentration of 10–20%. These crystals gave rise to good diffraction with good DNA density.

**Data collection and structure determination.** The diffraction data of I3C-soaked ZmMOC1 were collected by the in-house X-ray diffraction instrument (Rigaku). The diffraction data of HJ-free and HJ-bound NtMOC1 were all collected at Shanghai Synchrotron Research Facility (SSRF) on beamline BL17U or BL19U. The data were integrated and processed with the HKL2000 program suite and XDS packages[51]. Further data processing was carried out using CCP4 suit[52]. Crystal structure of ZmMOC1 was determined at a resolution of 2.5 Å. The structure of apo and HJ-bound NtMOC1 were solved by molecular replacement using ZmMOC1 as search template. All the structures were iteratively built with COOT[53] and refined with PHENIX program[54]. Data collection and structure refinement statistics are summarized in Supplementary Table 1. All figures were generated using the program PyMOL (http://www.pymol.org/).

**Electrophoretic mobility shift assay (EMSA).** Proteins were incubated with approximately 250 nM HJ substrates in a volume of 10 μl containing the binding buffer (50 mM Tris-HCl, pH 8.0, 1 mM DTT, 5 mM EDTA, 200 ng ml$^{-1}$ Heparin and 10% glycerol). After reaction for 30 min at 25 °C, the mixtures were resolved on 6% native acrylamide gels (37.5:1 acrylamide:bis-acrylamide) in 0.5× Tris-glycine buffer under an electric field of 15 V cm$^{-1}$ for 3 h. Gels were extracted and stained by GelRed, and visualized by Image Lab (Bio-Rad). The results are representative of at least three independent experiments.

**HJ cleavage assay.** The cleavage reaction (10 μl) was performed in the cutting buffer (50 mM Tris-HCl, pH 8.0, 1 mM DTT, 10 mM MgCl$_2$). The final HJ concentration is 250 nM, and protein concentrations are gradually 2-fold diluted from 1 μM. After 1 h reaction at 30 °C, the mixture was terminated by adding the termination buffer (50 mM Tris-Cl, pH 8.0, 3% SDS, 50 mM EDTA, Trypsin and pronase) and then incubated at 50 °C for 10 min. The reaction products were separated using 7% native PAGE. Gels were stained by GelRed, and visualized by Image Lab (Bio-Rad). The results are representative of at least three independent experiments.

**Cleavage site mapping.** The cleavage reaction (10 μl) was performed in the cutting buffer (50 mM Tris-Cl, pH 8.0, 1 mM DTT, 10 mM MgCl$_2$). The 5′-end of one of the DNA strands (1, 2, 3 or 4) was labeled with FAM. The final HJ concentration is 250 nM, and the protein concentration is 1 μM. The cleavage reaction was performed at 30 °C for 1 h, 5 μl of the digested HJ substrates were further mixed with 15 μl of the denaturing buffer (90% (v/v) formamide, 20 mM Tris-HCl, pH 8.0, 20 mM EDTA, pH 8.0, 0.05% (w/v) bromophenol blue and 0.05% xylene cyanol), and denatured by incubating at 98 °C for 10 min. Then the

reaction products were resolved on denaturing PAGE (23% acrylamide, 7 M urea, 1× TBE, 1% (w/v) APS and 0.1% TEMED). Fluorescent signals were detected by a scanner (Amersham Typhoon). To map the specific cleavage site, a series of 5′-FAM labeled oligos (12−17 nt) that include the homologous cores were used as markers.

**Size exclusion chromatography (SEC)**. To evaluate the molecular mass of NtMOC1 and the relative mutants, proteins were subjected to SEC analysis (Superdex-200 Increase 10/300, GE Healthcare) using an AKTA FPLC instrument (GE healthcare) in a buffer containing 25 mM Tris-HCl, pH 8.0, 500 mM NaCl. Fractions with the same elution volume from each injection were subjected to SDS-PAGE and visualized by Comassie blue staining. The elution volume of the individual injections could be used to assess their molecular masses.

**Analytical ultracentrifugation (AUC)**. The stoichiometry of MOC1 were investigated by AUC experiments, which was performed in a Beckman Coulter XL-I analytical ultracentrifuge using two-channel centrepieces. The proteins were in solutions containing 25 mM Tris-HCl, pH 8.0, 500 mM NaCl. Data were collected by absorbance detection at 18 °C for proteins at a concentration of 0.8–1 mg ml$^{-1}$ at a rotor speed of 45,000 r.p.m. The SV-AUC data were globally analyzed using the SEDFIT program and fitted to a continuous c(s) distribution model to determine the molecular mass.

**Reporting summary**. Further information on research design is available in the Nature Research Reporting Summary linked to this article.

## Data availability

The authors declare that all relevant data supporting the findings of this study are available within the article and its supplementary files or from the corresponding author upon reasonable request. The atomic coordinates and structure factors for the reported crystal structures have been deposited in the Protein Data Bank (PDB) with the accession codes 6KVN, 6KVO, 6LCM, 6LCT. The source data underlying Figs. 1b, 3d, e, and 5b, c and Supplementary Figs 3b, c, e, f, 5d, e, 7a, b, 10c, d, and 12a, b are provided as a Source Data file.

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

## Acknowledgements

We thank the staff of the BL17U1/BL19U1/BL19U2 beamline of the NCPSS at the Shanghai Synchrotron Radiation Facility for assistance during data collection; and research associates at the Center for Protein Research, Huazhong Agricultural University, for technical support. This work was supported by funds from the Ministry of Science and Technology of China (2018YFA0507700), the National Natural Science Foundation of China (31700203 for J.Y. and 31722017 for P.Y.), the Fork Ying-Tong Education Foundation (151021), the Fundamental Research Funds for the Central Universities (2662017PY031), and the China Postdoctoral Science Foundation (2017T100561).

## Author contributions

J.Y., S.H., and P.Y. designed all experiments. J.Y., S.H., and W.H. performed protein purification and crystallization. Z.G. determined all the structures. J.Y., S.H., and D.Z. performed the biochemical assays. All authors analyzed the data and contributed to paper preparation. J.Y. and P.Y. wrote the paper.

## Competing interests

The authors declare no competing interests.
