## [Peer Review File · Nature Communications]

Reviewers' comments:

Reviewer #1 (Remarks to the Author):

Junjie Yan, Sixing Hong, Zeyuan Guan, Wenjing He, Delin Zhang, and Ping Yin solved the crystal structure of *Nicotiana tabacum* MOC1 (NtMOC1) complexed with or free of Holliday junction (HJ) at a resolution of 2.0 Å and 2.5 Å, respectively. HJ is an intermediate of homologous genetic recombination. Although the authors do not mention in detail in this submitted paper, during DNA replication and DNA repair homologous recombination plays essential roles, but even under normal growing conditions, homologous recombination also causes the formation of complex DNA structures connected by HJs. Thus, the resolution of HJ is required for normal segregation of genomic DNA of prokaryotes and eukaryotes, and DNA of organelle nucleoids. Therefore, HJ resolvase would draw attentions of readers of various fields.

NtMOC1 is a member of sequence-dependent HJ resolvases. This group of HJ resolvases includes bacterial RuvC, budding yeast mitochondrial Cce1, fission yeast mitochondrial Ydc2 (SpCce1). The resolution of the crystal structure of NtMOC1-HJ complex is higher than those of the HJ resolvase-HJ complexes reported in the preceding publications including a sequence-specific HJ resolvase RuvC (ref. No. 39 in this paper) and a sequence-independent HJ resolvase GEN1 (Lilley 2017). The conclusions and explanations that the authors drew from their structures may not be very surprising. It is the authors' original claim that they characterize a base recognition loop (BR loop), and that residues from the BR loop intercalate into the bases, disrupt the C-G base pairing at the crossover and recognize the cytosine, providing the molecular bases of sequence-dependent HJ resolution by a resolvase. This claim appears to be similar to that described in a preceding publication describing RuvC (ref. No. 39). However, in the preceding paper, the authors' claims were based on their the molecular dynamics (MD) simulations and the crystal structure at the resolution of 3.4 Å. Because of the high resolution (2.0 Å) of the current structure, the authors of the current paper succeeded to explain more decisively and directly from the crystal structure the details of the sequence recognition by NtMOC1 as well as those of the binding of HJ and the cleavage mechanism. In addition, their conclusions and explanations are well supported biochemical experiments using appropriate amino-acid replacing mutant NtMOC1s with sufficient controls. Thus, I would like to support the publication of this paper in Nature Communications. Then, a question about supplementary Figure 7 is critical: are all structures of BR loops originally modeled by the authors of this paper? If even one of these structures was published previously, the publication of the current paper is no more supported. Although the authors described in the legend of this figure that "The follow three panels are the apo structures of EcRuvC, TtRuvC and Ydc2", I would like to advised that the authors clarify the originality of the materials shown in the supplementary Figure 7.

The followings are to be considered before publication:

1. In the discussion section, the authors criticize a preceding paper (ref. No. 39) reporting the crystal structures of the TtRuvC-HJ complex that "The catalytic residue Asp7 is far away from the scissile phosphates, revealing a not fully catalytic state of the complex." However, this comment is inappropriate and should be fixed. Under the reaction conditions both proteins and bound DNA have dynamic conformations and the differences between the current structural features and those of the cited paper may not be significant as discussed by the authors of the cited paper (ref. No. 39). It should be considered that the crystal structure described in this paper is just a snap-shot of dynamic structures under the reaction conditions, and that the preceding paper (ref. No. 39) well complements this weak point.
2. It appears that the authors do not pay attention whether the HJs that the authors used are fixed junctions or mobile ones. The HJ formed during homologous recombination moves along a stretch of a homologous sequence between parental DNAs (mobile HJ), but in many in vitro experiments, authors use fixed HJ in which the junction does not move at all. It is known that so called HJ resolvases would discriminate these two types of HJs. I would like to advise that the authors discuss about the relationship between the mobility and sequence specific and/or structure-specific nature of the HJ cleavage by NtMOC1.

3. The authors use two different sets of abbreviations to express amino acid-residues, single letter ones and three-letter ones. They should chose one of them throughout the paper.

Reviewer #2 (Remarks to the Author):

The manuscript by Yin, et al., describes the structure of the Holliday junction resolvase MOC1 from plant chloroplasts, as a protein alone and in complex with the DNA junction. The study shows how a loop in binding site of the protein undergoes a conformational shift in order to form a hydrogen bond to a cytosine base at the junction's non-crossing strand. This interaction was proposed to provide specificity for the junction sequence, which was further supported by resolvase and binding studies o mutations made to this region of the enzyme. Thus, the authors present convincing structural and biochemical evidence to support a unique mode of recognition for DNA junctions that promote sequence-specific recombination events. The work is thorough and, in general, well presented. The authors should, however, consider the following.

1. The authors show the electron density map of the DNA junction (Fig. 3) and of side chains around the catalytic center (Supplemental Fig. 4). However, the most interesting and important point of the paper is in the interactions of the protein with the DNA junction, particularly the putative hydrogen bonding interactions of the Asp side chain with the cytosine base. The authors should show the electron density in this region as a Fo-Fc difference omit map, where the important protein and DNA regions are omitted from the refinement process prior to phasing the electron density. This type of map provides the most unbiased demonstration of the structural details to support the authors' claims.

2. Although the structure demonstrates how the enzyme recognizes a static structure of a fixed junction, it may be useful to have a brief description for how the enzyme may contribute to a junction that would be capable of migrating.

3. The manuscript can use with some editing for the use of the English language.

Reviewer #3 (Remarks to the Author):

Yan et al present the crystal structure of the RuvC domain of the chloroplast Holliday junction (HJ) resolving enzyme MOC1 from *Nicotiana tabacum*, both on its own and as a complex with a four-way DNA junction, with a view to understanding the molecular basis of the sequence-specificity of this enzyme. Overall, their findings confirm those of the study of the MOC1 ortholog from *Zea mays* (just published by Lin et al in *Nature Chemical Biology*: <https://doi.org/10.1038/s41589-019-0377-4>), which the authors will need to acknowledge and discuss.

They show that, like all HJ-resolving enzymes characterised so far, MOC1 functions as a dimer, binds Holliday junctions predominantly through electrostatic interactions with the phosphate backbone and disrupts base stacking at the branchpoint. In contrast with the Tt RuvC structures published by the Nowotny lab, they observe sequence-specific contacts with one nucleobase (the cytosine immediately 5' to the cleavage site), whose base-pairing partner has been flipped out of the DNA helix. Their main structural conclusions are backed-up by the biochemical characterisation of a series of mutants.

Overall, the results are well presented, however information is missing in places for the experiments to be fully reproducible and some of the claims are not fully supported by the data presented. This needs to be addressed before publication. Please see specific points below.

Main points

1. Cleavage assays - This is my main concern.

To analyse the products of the reaction, the authors heat-denature their samples in the presence of formamide, then run them on a native polyacrylamide gel (p.13, line 303-306). Thus, what they observe in the gel is single-stranded DNA of different sizes (cleaved vs uncleaved) rather than HJ vs nicked duplex as suggested by the diagrams alongside the gels in all of the corresponding figures (1b, 3e, 5c, sup.2e, sup.4d, sup.6b). The other consequence of this method choice is that what they observe is not HJ resolution (i.e. the bilateral cleavage event that results in the formation of two nicked duplexes) as stated in some of the figure legends but merely cleavage (which could be unilateral or bilateral). This needs to be corrected as it is quite misleading. Alternatively, the authors could repeat these experiments and run their samples on native gels without denaturing them to observe HJ resolution rather than cleavage. Moreover, the position of the cleavage sites has not been mapped. While this position is known from previous work (ref. 19) for junction X2 (CCGG), the possibility of branch migration through the homologous core makes it uncertain for the other sequences used in this study. Additionally, since the DNA is detected via post-electrophoretic gel-staining rather than labelling of a single oligonucleotide, the detected shorter oligonucleotides could result from cleavage of any of the four strands. The authors should remove the arrows denoting cleavage site positions in the text (line 83) and in fig.1a for junctions II and III. In conclusion, the experiments presented here are not sufficient to characterise the sequence-specificity of MOC1 orthologs.

2. Information missing in the methods section

- DNA annealing buffer

The composition of the buffer used to anneal the HJ is not specified and hence the exact composition of the crystallisation drops is unknown.

- structure resolution

The authors state (line 284-287) that they "firstly determined the crystal structure of MOC1 from *Zea mays* (ZmMOC1) using I3C commercial kit via single-wavelength anomalous diffraction (SAD) method", then used this structure to solve the presented structures of NtMOC1 by molecular replacement. They should provide more information. What were the crystallisation conditions for ZmMOC1? Was it co-crystallised with I3C or were crystals obtained independently and then soaked with I3C? What was the quality of the obtained structure? And why is it not described nor shown?

- binding assays

The protein concentration is not specified, nor is the sequence of the HJ. This second point is important since the binding buffer contains 5 mM MgCl₂, which could induce cleavage if a cognate sequence was used.

- cleavage assays

The sequence of the HJ used is not specified (with the exception of the assays shown in fig. 1)

3. Identification of a Mg²⁺ ion in the active site (line 133 and figure legends)

The crystallisation conditions don't appear to include any magnesium ions (unless they are present in the annealing buffer - see point 2) and the authors present no justification for this assignment. Do the dotted lines in fig. 3c and sup. fig. 4b represent inner sphere contacts? Are the distances consistent with magnesium coordination or could this be a monovalent ion (the crystallisation drops contain sodium and lithium)? If this claim can't be substantiated, then the text and figure legends should be amended.

4. dimer interface

The authors show that mutations in the dimer interface, which prevent dimer formation in solution, severely reduce HJ-binding and abolish cleavage. This is an interesting result and to my knowledge no such mutant of RuvC homologs has been described so far. The authors could explore this further (for example can the activity be rescued at higher protein concentration?) and discuss why

these mutations have such a drastic effect (although a dimer is needed for productive HJ resolution, one could envisage the formation of an active dimeric complex by sequential binding of two monomers).

5. non-cognate sequence

The HJ used in the complex structure contains a CATG core (shown in sup. Fig. 5), which NtMOC1 does not cleave (see fig. 1). Hence the sequence-specific interaction observed in this structure, which is presumably required for cleavage to proceed, is not sufficient. The authors should make this clear in their discussion and could compare their structure to the ZmMOC1 complex structure presented by Lin et al, which contains a cognate sequence (CCGG core).

Minor points

Line 19: please correct the typo "MOC1 cleave HJ" (MOC1, which cleaves HJ...)

Line 42: "nuclear GEN1". Although GEN1 is encoded in the nuclear genome, it is predominantly localised in the cytoplasm (see more recent papers from the West lab).

Line 54: "we characterized MOC1s cleaved HJ". This makes no sense. Please rephrase.

Line 90: the authors reference four examples of "known" homodimeric resolvases (RuvC, T4 endonuclease VII, T7 endonuclease I and Ydc2). What about Rusa, RecU, Hjc, Hje, GEN1?

Line 124: please rephrase the description of the HJ conformation. "Square-planar" suggests 90° angles between the duplex arms, and "stacked-X" suggests basepairs are stacked across the branchpoint. Neither applies to this structure.

Line 136: please correct the typo "Asp230". Should be Asp253.

Line 144-146: please rephrase. The position of the scissile phosphate cannot be known a priori since the branchpoint could take three distinct positions due to homology. Moreover, the CATG core is not cleaved by NtMOC1, and experiments in ref. 19 map the cleavage sites of a HJ containing a CCGG core. However its position can be inferred from its proximity to the enzyme's active site in the structure.

Line 156, 164: the contact with Lys225 appears to be through its main chain amine (fig. 4d), hence it should be anticipated that mutating this residue won't affect activity.

Line 158: "non-exchanging DNA strands". Should be "exchanging DNA strands".

Line 259-260: replace "annealed in boiled water" with "incubated in boiled water" and provide the composition of the annealing buffer.

Line 268: the crystallised protein is a quadruple mutant. How does its activity compare with that of the wild type enzyme?

Fig. 4: please reposition the yellow rectangles so that they actually encompass the area depicted in fig. 4b.

Sup. Fig. 2: please indicate the theoretical MW of a monomer.

Response to reviewers' comments:

Reviewer #1:

*Junjie Yan, Sixing Hong, Zeyuan Guan, Wenjing He, Delin Zhang, and Ping Yin solved the crystal structure of *Nicotiana tabaccum* MOC1 (NtMOC1) complexed with or free of Holliday junction (HJ) at a resolution of 2.0 Å and 2.5 Å, respectively. HJ is an intermediate of homologous genetic recombination. Although the authors do not mention in detail in this submitted paper, during DNA replication and DNA repair homologous recombination plays essential roles, but even under normal growing conditions, homologous recombination also causes the formation of complex DNA structures connected by HJs. Thus, the resolution of HJ is required for normal segregation of genomic DNA of prokaryotes and eukaryotes, and DNA of organelle nucleoids. Therefore, HJ resolvase would draw attentions of readers of various fields.*

NtMOC1 is a member of sequence-dependent HJ resolvases. This group of HJ resolvases includes bacterial RuvC, budding yeast mitochondrial Cce1, fission yeast mitochondrial Ydc2 (SpCce1). The resolution of the crystal structure of NtMOC1-HJ complex is higher than those of the HJ resolvase-HJ complexes reported in the preceding publications including a sequence-specific HJ resolvase RuvC (ref. No. 39 in this paper) and a sequence-independent HJ resolvase GEN1 (Lilley 2017).

The conclusions and explanations that the authors drew from their structures may not be very surprising. It is the authors' original claim that they characterize a base recognition loop (BR loop), and that residues from the BR loop intercalate into the bases, disrupt the C-G base pairing at the crossover and recognize the cytosine, providing the molecular bases of sequence-dependent HJ resolution by a resolvase. This claim appears to be similar to that described in a preceding publication describing RuvC (ref. No. 39). However, in the preceding paper, the authors' claims were based on the molecular dynamics (MD) simulations and the crystal structure at the resolution of 3.4 Å. Because of the high resolution (2.0 Å) of the current structure, the authors of the current paper succeeded to explain more decisively and directly from the crystal structure the details of the sequence recognition by NtMOC1 as well as those of the binding of HJ and the cleavage mechanism.

In addition, their conclusions and explanations are well supported biochemical experiments using appropriate amino-acid replacing mutant NtMOC1s with sufficient controls. Thus, I would like to support the publication of this paper in Nature Communications. Then, a question about supplementary Figure 7 is critical: are all structures of BR loops originally modeled by the authors of this paper? If even one of these structures was published previously, the publication of the current paper is no more supported. Although the authors described in the legend of this figure that "The follow three panels are the apo structures of EcRuvC, TtRuvC and Ydc2", I would like to advised that the authors clarify the originality of the materials shown in the supplementary Figure 7.

We truly thank the referee for his/her support for the publication of our work in Nature Communications.

We appreciate the critical comment on Supplementary Figure 7. The crystal structures of EcRuvC (PDB ID: 1hjr), TtRuvC (PDB ID: 4ep4) and Ydc2 (PDB ID: 1kcf) shown in the figure have been previously reported. And sequence-dependent HJ resolution has been demonstrated for these enzymes. However, the underlying molecular basis of the selectivity and the potential key roles of this loop remain largely elusive. To further clarify the originality of the potential BR loop of these resolvase, we have reorganized the figure (dividing the original Figure S7a into two parts, please see Supplementary Figure 14a, b) and rephrased the figure legend in the revised manuscript. We hope this modified version will be clear enough to be understood.

The followings are to be considered before publication:

1) In the discussion section, the authors criticize a preceding paper (ref. No. 39) reporting the crystal structures of the TtRuvC-HJ complex that “The catalytic residue Asp7 is far away from the scissile phosphates, revealing a not fully catalytic state of the complex.” However, this comment is inappropriate and should be fixed. Under the reaction conditions both proteins and bound DNA have dynamic conformations and the differences between the current structural features and those of the cited paper may not be significant as discussed by the authors of the cited paper (ref. No. 39). It should be considered that the crystal structure described in this paper is just a snap-shot of dynamic structures under the reaction conditions, and that the preceding paper (ref. No. 39) well complements this weak point.

We appreciate this insightful suggestion. We agree with the referee that the crystal structure of NtMOC1-HJ obtained by us and TtRuvC-HJ by the other group are just snapshots of the dynamic complex structures. We have deleted this sentence “The catalytic residue Asp7 is far away from the scissile phosphates, revealing a not fully catalytic state of the complex” in the revised manuscript.

2) It appears that the authors do not pay attention whether the HJs that the authors used are fixed junctions or mobile ones. The HJ formed during homologous recombination moves along a stretch of a homologous sequence between parental DNAs (mobile HJ), but in many in vitro experiments, authors use fixed HJ in which the junction does not move at all. It is known that so called HJ resolvases would discriminate these two types of HJs. I would like to advise that the authors discuss about the relationship between the mobility and sequence specific and/or structure-specific nature of the HJ cleavage by NtMOC1.

We appreciate this insightful comment. In a previous study, the authors characterized that MOC1 could cleave mobile HJs, using either X12 HJ (with a 12-bp homologous core) or X2 HJ (with a 2-bp homologous core) as substrates (Kobayashi et al., 2017, *Science*). In this paper, we used bimobile

X2 HJ with 2-bp homologous cores for crystallization and *in vitro* binding and cleavage assays. To exhibit the bimobility of the HJs used in this paper, we prepared a new figure in Supplementary Figure 2 in the revised manuscript (Thus, all subsequent supplementary figures have been renumbered).

Supplementary Figure 2. Bimobility of the X2 HJ substrates. **a**, Schematic drawing of the X2 (CCGG) substrate. X2 HJ was prepared by annealing 4 oligos with 2-bp homologous core sequence. **b**, Branch migration of the X2 (CCGG). X2 harbors 2-bp homologous core sequence (gray background) that exhibits branch migration of two steps within the core sequence. Other X2 substrates used in this study exhibit identical bimobile characteristics as the X2 (CCGG).

HJs are highly dynamic structures that can undergo conformer exchange and branch migration. From the reported crystal structures of resolvases (T4, T7, TtRuvC) in complex with HJ and NtMOC1-HJ complex, we can learn that

resolvase interact with the HJ DNA backbone extensively. Recently, using single-molecule fluorescence resonance energy transfer (smFRET), a study indicated that the HJs remain dynamic in the presence of resolving enzymes (as exemplified by T7 endo I, RuvC, GEN1, and hMus81-Eme1) (Zhou et al., 2019, *Nature Chemical Biology*). The dimeric enzymes use their multivalent interactions with the HJ, going through a partially dissociated intermediate, contributing to an unencumbered HJ dynamics. The study demonstrated that there is coordination among junction resolution, conformer exchange and branch migration (Zhou et al., 2019, *Nature Chemical Biology*).

Similarly, we speculate that the presence of NtMOC1 might not restrain the dynamics of the HJ, which can undergo branch migration, extending or shortening the length of DNA heteroduplex. Once the cognate sequence of CCGG (or CGCG) moves to the branch point, residues Y180 and D183 from the dynamic BR loop of NtMOC1 could penetrate into the junction and disrupt the C-G base pairs. Meanwhile, the disrupted cytosine is recognized by D183, generating a catalytic HJ configuration, thus facilitating the hydrolysis of the phosphodiester bond by the catalytic tetrad of NtMOC1. We have discussed this possibility in the discussion part (lines 210-228).

3) The authors use two different sets of abbreviations to express amino acid-residues, single letter ones and three-letter ones. They should choose one of them throughout the paper.

Point accepted. We have modified the abbreviations and use single letters to represent the amino acid residues throughout the paper.

We thank this reviewer for his/her constructive comments.

Reviewer #2:

The manuscript by Yin, et al., describes the structure of the Holliday junction resolvase MOC1 from plant chloroplasts, as a protein alone and in complex with the DNA junction. The study shows how a loop in binding site of the protein undergoes a conformational shift in order to form a hydrogen bond to a cytosine base at the junction's non-crossing strand. This interaction was proposed to provide specificity for the junction sequence, which was further supported by resolvase and binding studies of mutations made to this region of the enzyme. Thus, the authors present convincing structural and biochemical evidence to support a unique mode of recognition for DNA junctions that promote sequence-specific recombination events. The work is thorough and, in general, well presented.

We would like to thank the referee for his/her remark that our work is thorough and well presented.

The authors should, however, consider the following.

1) The authors show the electron density map of the DNA junction (Fig. 3) and of side chains around the catalytic center (Supplemental Fig. 4). However, the most interesting and important point of the paper is in the interactions of the protein with the DNA junction, particularly the putative hydrogen bonding interactions of the Asp side chain with the cytosine base. The authors should show the electron density in this region as a Fo-Fc difference omit map, where the important protein and DNA regions are omitted from the refinement process prior to phasing the electron density. This type of map provides the most unbiased demonstration of the structural details to support the authors' claims.

We appreciate this advice. We have prepared a new figure with 2Fo-Fc map (1.0 σ , lightblue) and Fo-Fc map ($\pm 3.0 \sigma$, purple and red) superimposed to show the electron density of Y180 and D183 and the related bases (Supplementary Figure 13). The high quality of the electron density map and the omit map reveals the fidelity of the crucial roles of Y180 and D183 involved in the disruption of the C-G base pair at the branch point and the sequence-specific recognition of cytosine by D183 of NtMOC1.

Supplementary Figure 13. Electron density map of base recognition residues and the bases at the crossover of the NtMOC1-HJ complexes. **a**, The superimposed 2Fo-Fc (1.0σ , lightblue) and Fo-Fc ($\pm 3.0 \sigma$, purple and red) map of key elements from the complex structure composed of non-cognate CATG core sequence. **b**, The superimposed 2Fo-Fc (1.0σ , lightblue) and Fo-Fc ($\pm 3.0 \sigma$, purple and red) map of key elements from the complex structure composed of cognate CCGG core sequence. Residues and bases are labeled.

2) *Although the structure demonstrates how the enzyme recognizes a static structure of a fixed junction, it may be useful to have a brief description for how the enzyme may contribute to a junction that would be capable of migrating.*

We appreciate this insightful comment. A similar issue is also raised by referee#1. In fact, the HJs used in this study (for crystallization and biochemical assays including binding and cleavage) are all bimobile. We have added a new figure to demonstrate this point in Supplementary Figure 2 in the revised manuscript.

Supplementary Figure 2. Bimobility of the X2 HJ substrates. **a**, Schematic drawing of the X2 (CCGG) substrate. X2 HJ was prepared by annealing 4 oligos with 2-bp homologous core sequence. **b**, Branch migration of the X2 (CCGG). X2 harbors 2-bp homologous core sequence (gray background) that exhibits branch migration of two steps within the core sequence. Other X2 substrates used in this study exhibit identical bimobile characteristics as the X2 (CCGG).

HJs are highly dynamic structures that can undergo conformer exchange and branch migration. From the reported crystal structures of resolvases (T4, T7, TtRuvC) in complex with HJ and NtMOC1-HJ complex, we can learn that resolvase interact with the HJ DNA backbone extensively. Recently, using single-molecule fluorescence resonance energy transfer (smFRET), a study indicated that the HJs remain dynamic in the presence of resolving enzymes (as exemplified by T7 endo I, RuvC, GEN1, and hMus81-Eme1) (Zhou et al.,

2019, *Nature Chemical Biology*). The dimeric enzymes use their multivalent interactions with the HJ, going through a partially dissociated intermediate, contributing to an unencumbered HJ dynamics. The study demonstrated that there is coordination among junction resolution, conformer exchange and branch migration (Zhou et al., 2019, *Nature Chemical Biology*).

Similarly, we speculate that the presence of NtMOC1 might not restrain the dynamics of the HJ, which can undergo branch migration, extending or shortening the length of DNA heteroduplex. Once the cognate sequence of CCGG (or CGCG) moves to the branch point, residues Y180 and D183 from the dynamic BR loop of NtMOC1 could penetrate into the junction and disrupt the C-G base pairs. Meanwhile, the disrupted cytosine is recognized by D183, generating a catalytic HJ configuration, thus facilitating the hydrolysis of the phosphodiester bond by the catalytic tetrad of NtMOC1. We have discussed this possibility in the discussion part (lines 210-228).

3) The manuscript can use with some editing for the use of the English language.

Point accepted. We have polished our revised manuscript using English language editing service from Springer Nature (order: DL159CG5).

We thank this reviewer for his/her constructive comments.

Reviewer #3:

Yan et al present the crystal structure of the RuvC domain of the chloroplast Holliday junction (HJ) resolving enzyme MOC1 from Nicotiana tabaccum, both on its own and as a complex with a four-way DNA junction, with a view to understanding the molecular basis of the sequence-specificity of this enzyme. Overall, their findings confirm those of the study of the MOC1 ortholog from Zea mays (just published by Lin et al in Nature Chemical Biology: <https://doi.org/10.1038/s41589-019-0377-4>), which the authors will need to acknowledge and discuss.

They show that, like all HJ-resolving enzymes characterised so far, MOC1 functions as a dimer, binds Holliday junctions predominantly through electrostatic interactions with the phosphate backbone and disrupts base stacking at the branchpoint. In contrast with the Tt RuvC structures published by the Nowotny lab, they observe sequence-specific contacts with one nucleobase (the cytosine immediately 5' to the cleavage site), whose base-pairing partner has been flipped out of the DNA helix. Their main structural conclusions are backed-up by the biochemical characterisation of a series of mutants.

Overall, the results are well presented, however information is missing in places for the experiments to be fully reproducible and some of the claims are not fully supported by the data presented. This needs to be addressed before publication. Please see specific points below.

We truly thank the referee for his/her remark that our work is well presented.

Main points

1) Cleavage assays - This is my main concern

To analyse the products of the reaction, the authors heat-denature their samples in the presence of formamide, then run them on a native polyacrylamide gel (p.13, line 303-306). Thus, what they observe in the gel is single-stranded DNA of different sizes (cleaved vs uncleaved) rather than HJ vs nicked duplex as suggested by the diagrams alongside the gels in all of the corresponding figures (1b, 3e, 5c, sup.2e, sup.4d, sup.6b). The other consequence of this method choice is that what they observe is not HJ resolution (i.e. the bilateral cleavage event that results in the formation of two nicked duplexes) as stated in some of the figure legends but merely cleavage (which could be unilateral or bilateral). This needs to be corrected as it is quite misleading. Alternatively, the authors could repeat these experiments and run their samples on native gels without denaturing them to observe HJ resolution rather than cleavage.

Thanks for this very insightful comment. We initially used both native PAGE and denaturing PAGE (using formamide and high temperature to denature the MOC1-HJ sample and disrupt the base pairs) to detect the cleavage activity of MOC1. However, in the preparation of this manuscript, we presented the results that are all using native PAGE without formamide to denature the sample. We have modified the demonstrations of the HJ cleavage assay in the

Method section and removed the sentence “The digested HJ substrates were mixed with 15 μ l of the denaturing buffer (90% (v/v) formamide, 20 mM Tris-Cl, pH 8.0, 20 mM EDTA, pH 8.0, 0.05% (w/v) bromophenol blue and 0.05% xylene cyanol”. The denaturing buffer was used to denature the HJ samples in the cleavage site mapping assay in the revised manuscript.

Moreover, the position of the cleavage sites has not been mapped. While this position is known from previous work (ref. 19) for junction X2 (CCGG), the possibility of branch migration through the homologous core makes it uncertain for the other sequences used in this study. Additionally, since the DNA is detected via post-electrophoretic gel-staining rather than labelling of a single oligonucleotide, the detected shorter oligonucleotides could result from cleavage of any of the four strands. The authors should remove the arrows denoting cleavage site positions in the text (line 83) and in fig.1a for junctions II and III.

In conclusion, the experiments presented here are not sufficient to characterise the sequence-specificity of MOC1 orthologs.

Thanks for this advice. In the revised manuscript, we have mapped the cleavage site for the junction of X2 (CATG) and X2 (CGCG) by denaturing PAGE using 5'-FAM labeled oligo sequences as DNA markers. The results revealed strands 2 and 4 of the junctions are cleavable. And MOC1 introduces nicks symmetrically into the sequence 5'-C \downarrow A-3' (\downarrow indicate the cutting site) for X2 (CATG), and 5'-C \downarrow G-3' for X2 (CGCG). We have demonstrated these results in the main text and added the results in Supplementary Figure 3. The slightly different cleavage results for X2 (CATG) and X2 (CGCG) obtained by us and the other group might be caused by different reaction systems applied, especially the MOC1 protein concentrations (we used protein concentration in μ M levels, much higher than the previous study), considering the cleavage activity of MOC1 against X2 (CATG) and X2 (CGCG) are obviously lower than that for X2 (CCGG). These biochemical results revealed MOC1s introduce nicks symmetrically after a cytosine at the branch point, which is further validated by the crystal structures of NtMOC1-HJ complex in this study and ZmMOC1-HJ complex in another study (Lin et al, 2019, *Nature Chemical Biology*).

Supplementary Figure 3. Cleavage site mapping for junction X2 (CATG) and X2 (CGCG). **a,d**, Schematic drawing of the X2 (CATG) and X2 (CGCG) substrates. Arrows indicate the cleavage site revealed by the following mapping assays. **b,e**, Strands 2 and 4 of the X2 (CATG) and X2 (CGCG) are cleavable DNA strands. The 5' end of one of the DNA strands (1, 2, 3 or 4) was labeled with FAM. The concentration of HJ in each reaction is 250 nM. Protein concentration is 1000 nM. AtMOC1 was used as a representative for the mapping assay. The reaction products were resolved by denaturing PAGE. **c,f**, Cleavage site mapping for strands 2 and 4 of the X2 (CATG) and X2 (CGCG). The reaction products were resolved by denaturing PAGE and mapped by 5'-FAM labeled oligo sequences. **g**, 5'-FAM labeled oligo nucleotide sequences used as marker for cleavage site mapping. Source data are provided as a Source Data file.

2) Information missing in the methods section

- DNA annealing buffer

The composition of the buffer used to anneal the HJ is not specified and hence the exact composition of the crystallisation drops is unknown.

Thanks for this advice. We have added the composition of the DNA annealing buffer (25 mM Tris-HCl, pH 8.0, 150 mM NaCl) in the method (Line 296).

- structure resolution

The authors state (line 284-287) that they “firstly determined the crystal structure of MOC1 from *Zea mays* (ZmMOC1) using I3C commercial kit via single-wavelength anomalous diffraction (SAD) method”, then used this structure to solve the presented structures of NtMOC1 by molecular replacement. They should provide more information. What were the crystallisation conditions for ZmMOC1? Was it co-crystallised with I3C or were crystals obtained independently and then soaked with I3C? What was the quality of the obtained structure? And why is it not described nor shown?

Thanks for this advice. We have demonstrated the crystallization condition for ZmMOC1 “The rodlike crystals of ZmMOC1 (T107-V280) were grown in a well buffer containing 14% PEG3350, 0.15 M NH₄Cl and grew to full size within 3 days. High-quality ZmMOC1 crystals were soaked in the reservoir solution plus 0.1 M 5-amino-2,4,6-triiodoisophthalic acid (I3C, Molecular Dimensions) for 2 hours” (Lines 304-309). The diffraction data were collected by the in-house X-ray diffraction instrument (Rigaku). We determined the crystal structure of HJ-free ZmMOC1 (T107-V280) via iodide-based single-wavelength anomalous diffraction (I-SAD) at a refined resolution of 2.5 Å (Lines 333-335). We added the crystal structure of ZmMOC1 in the supplementary Figure 4 in the revised manuscript. Structure superimposition of the iodide-phased ZmMOC1 with the recently published selenium-phased ZmMOC1 revealed a similar architecture with a RMSD of 0.45 Å.

Supplementary Figure 4. Crystal structure of ZmMOC1. **a**, Electron density map of I3C compounds in ZmMOC1. The map is contoured at 3 σ (pink). The equilateral triangle formed by the I atom is clearly visible. **b**, Crystal structure of ZmMOC1 (T107-V280) determined in this study. Protomer A and B are colored in wheat and cyan, respectively. The same color scheme is used for the following panel of ZmMOC1. **c**, Structural superimposition of NtMOC1 with ZmMOC1. Protomer A and B of NtMOC1 are colored in slate and yellow, respectively. Structural alignment reveals an RMSD value of 0.919 Å over 257 C α .

- binding assays

The protein concentration is not specified, nor is the sequence of the HJ. This second point is important since the binding buffer contains 5 mM MgCl₂, which could induce cleavage if a cognate sequence was used.

Thanks for this advice. We have added the protein gradient concentrations in each figure legend. And the sequences of the HJ are deposited in the Supplementary Table 3. In practice, the binding buffer does not contain MgCl₂. We apologize for the typos and have modified this error in the revised manuscript.

- cleavage assays

The sequence of the HJ used is not specified (with the exception of the assays shown in fig. 1)

Thanks for this advice. We have indicated the specific HJs used in each cleavage assays in the figure legends.

3) Identification of a Mg²⁺ ion in the active site (line 133 and figure legends)

The crystallisation conditions don't appear to include any magnesium ions (unless they are present in the annealing buffer - see point 2) and the authors present no justification for this assignment. Do the dotted lines in fig. 3c and sup. fig. 4b represent inner sphere contacts? Are the distances consistent with magnesium coordination or could this be a monovalent ion (the crystallisation drops contain sodium and lithium)? If this claim can't be substantiated, then the text and figure legends should be amended.

Thanks for this advice. From the previous study and our cleavage assay, we are aware of the necessity of magnesium (or manganese) ions for the MOC1's cleavage activity. Prior to crystallization, we have added a final concentration of 1 mM MgCl₂ to the protein sample. Thus, we observed the magnesium ions in the NtMOC1-HJ complex. We have added this demonstration in the crystallization section of Method in the revised manuscript (lines 304-306 and 313-314).

4) dimer interface

The authors show that mutations in the dimer interface, which prevent dimer formation in solution, severely reduce HJ-binding and abolish cleavage. This is an interesting result and to my knowledge no such mutant of RuvC homologs has been described so far. The authors could explore this further (for example can the activity be rescued at higher protein concentration?) and discuss why these mutations have such a drastic effect (although a dimer

is needed for productive HJ resolution, one could envisage the formation of an active dimeric complex by sequential binding of two monomers).

Thanks for this insightful suggestion. We have prepared fresh protein samples and increased the protein concentration (as high as 8 μ M, 8-fold concentration than the previous assays) to perform the HJ binding and cleavage assay for these NtMOC1 mutants with impaired dimerization. The AUC and SEC assays reveal that NtMOC1 (G200E/A204E) exhibits as a monomer compared with the wild-type NtMOC1. From the SEC assay, the single mutation of G200E or A204E also displays impaired dimerization. Though these mutants retain some HJ binding activity, the HJ resolution activity is strikingly diminished (Supplementary Figure 5). Even though, as the protein quantity increases to a high concentration, the HJ resolution activity of these mutants (G200E/A204E, G200E, and A204E) can be partially restored. This might be due to a potentially sequential binding of monomeric NtMOC1 that partially mimic a functional dimeric MOC1 as the reviewer postulated. These results reveal that the dimeric state of the enzyme is important for the efficient HJ cleavage.

We speculated that there are several reasons for the inefficient HJ cleavage by the monomeric MOC1. (1) From the complex structures of NtMOC1-HJ and ZmMOC1-HJ, each promoter of dimeric MOC1 contacts only one continuous strand, and dimeric NtMOC1 is able to distort the HJ in a favorable configuration for efficient cleavage. However, the monomeric NtMOC1 is less effective to distort the HJ in a correct configuration, resulting in a poor HJ cleavage activity. (2) The two symmetrical nicks on the HJ are occurred simultaneously or sequentially, suggesting coordination between the two protomers of the dimeric enzyme. But there is a lack of coordination for the monomeric MOC1. (3) Moreover, we cannot exclude the possibility that the DNA binding mode for the monomeric NtMOC1 might be different from the productive dimeric NtMOC1, leading to a less efficient HJ resolution. (4) The glutamate-substitution of glycine/alanine of NtMOC1 not only disrupt the dimer formation, but also lead to a potential electrostatic repulsion between the monomers, probably hindering the efficiently sequential binding of the two monomers. Collectively, all these defects lead to an inefficient HJ cleavage for the monomeric MOC1, suggesting MOC1 functions as a dimer.

Supplementary Figure 5. Disruption of the NtMOC1 dimerization affects its HJ resolution activity. d, HJ-binding activity of the mutants with impaired dimerization. **e**, HJ cleavage activity of the mutants with impaired dimerization. For the HJ binding and cleavage assay, X2 (CCGG) was used as the substrates. The final concentration of X2 (CCGG) in each lane is 250 nM. Six gradients with increasing concentrations (0, 500, 1000, 2000, 4000, and 8000 nM) were applied for each protein sample. The reactions were resolved by native PAGE and visualized by GelRed staining.

5) *non-cognate sequence*

The HJ used in the complex structure contains a CATG core (shown in sup. Fig. 5), which NtMOC1 does not cleave (see fig. 1). Hence the sequence-specific interaction observed in this structure, which is presumably required for cleavage to proceed, is not sufficient. The authors should make this clear in their discussion and could compare their structure to the ZmMOC1 complex structure presented by Lin et al, which contains a cognate sequence (CCGG core).

Thanks for this advice. Besides the NtMOC1-HJ complex presented in the initial submitted manuscript, we also determined the crystal structure of a cleavage inactive NtMOC1 mutant (D116A/E175A/D253A/E258A) in complex with HJ with cognate CCGG core sequence. The two complex structures exhibit nearly identical overall folds. Superimposition of the two HJs revealed the CATG and the CCGG core at the crossover are well aligned, where the C-G base pairs at the crossover are all disrupted. In both structures, we observed specific interactions between the disrupted cytosine (C9) and D183 of NtMOC1. We have included this complex structure (composed of a HJ with CCGG core) in the revised manuscript, and structure comparisons of these complexes are provided in Supplementary Figure 8 and 13. In addition, we also compared the NtMOC1-HJ with ZmMOC1-HJ complex. The two structures show nearly identical spatial configuration, where the HJs annealed either by four oligos (in our study) or two T-loop oligos (by Lin group) could be well aligned. Moreover, key residues Y180 and D183 in the BR loop of NtMOC1 and the correlated bases at the crossover can be well aligned with the corresponding residues F179 and D182 in the BR motif of ZmMOC1 and the related bases at the crossover. We have added the structure comparison of NtMOC1-HJ with ZmMOC1-HJ in Supplementary Figure 15 and discussed these results in the discussion (Lines 251-261).

Supplementary Figure 8. Structure comparison of NtMOC1 in complex with HJs with cognate sequence or non-cognate sequence. a,b, Schematic drawing of HJs crystallized in the complex structure with either non-cognate CATG core sequence (**a**) or cognate CCGG core sequence (**b**). The 2 bp homologous core sequences are highlighted in gray box. **c**, Structure superimposition of NtMOC1-HJ complexes. The colored cartoon indicates the complex structure of NtMOC1 and HJ with non-cognate CATG core sequence. The gray cartoon indicates the complex structure of the cleavage inactive NtMOC1 mutant (D116A/E175A/D253A/E258A) and HJ with cognate CCGG core sequence. **d**, Structure alignment of HJs with CATG and CCGG cores. The colored cartoon indicates the HJ with CATG core, whereas gray cartoon indicates the HJ with CCGG core. The CATG and CCGG cores at the branch point are labeled, and the nucleotides can be well aligned. Scissors indicate the cleavage site.

Supplementary Figure 13. Electron density map of base recognition residues and the bases at the crossover of the NtMOC1-HJ complexes. a, The superimposed 2Fo-Fc (1.0σ , lightblue) and Fo-Fc ($\pm 3.0 \sigma$, purple and red) map of key elements from the complex structure composed of non-cognate CATG core sequence. **b,** The superimposed 2Fo-Fc (1.0σ , lightblue) and Fo-Fc ($\pm 3.0 \sigma$, purple and red) map of key elements from the complex structure composed of cognate CCGG core sequence. Residues and bases are labeled.

Supplementary Figure 15. Structural comparisons of NtMOC1-HJ and ZmMOC1-HJ complex. **a**, Structural comparison of iodide-phased ZmMOC1 (by our study) with selenium-phased ZmMOC1 (PDB ID: 6IS9). For clarity, residues 86-95 from protomer A of 6IS9 are omitted. **b**, Structural superimposition of the complex structure of NtMOC1-HJ and ZmMOC1-HJ. **c**, Structural alignment of four oligos annealed HJ with two oligos annealed (with a T-loop) HJ in the complex structures. The magenta oval highlights the disrupted cytosine at the crossover of both HJs. The arrow indicates a T-loop. **d**, Structural alignment of the crucial residues and the bases involved in the sequence-specific HJ resolution. The colored cartoon represents the structure of NtMOC1-HJ determined in this study. The gray cartoon represents the structure of ZmMOC1-HJ determined by Lin group (PDB ID: 6IS8). Scissors indicate the scissile phosphates.

Minor points

6) Line 19: please correct the typo “MOC1 cleave HJ” (MOC1, which cleaves HJ...)

Point accepted. We have modified this typo.

7) Line 42: “nuclear GEN1”. Although GEN1 is encoded in the nuclear genome, it is predominantly localised in the cytoplasm (see more recent papers from the West lab).

Point accepted. We have learned that the majority of GEN1 are localized in the cytoplasm with an estimation of approximately 80%, and only 20% in the

nuclear fraction (Ying Wai Chan and Stephen C. West, 2014, *Nature Communications*). We have rephrased the sentence as “the nucleus-encoded Gen1 and its orthologs have been broadly investigated” (Lines 38-39).

8) Line 54: “we characterized MOC1s cleaved HJ”. This makes no sense. Please rephrase.

Point accepted. We have removed this sentence in the revised manuscript.

9) Line 90: the authors reference four examples of “known” homodimeric resolvases (*RuvC*, *T4* endonuclease VII, *T7* endonuclease I and *Ydc2*). What about *RusA*, *RecU*, *Hjc*, *Hje*, *GEN1*?

Thanks for this suggestion. We have added these dimeric resolvases to the reference list in the main text.

10) Line 124: please rephrase the description of the HJ conformation. “Square-planar” suggests 90° angles between the duplex arms, and “stacked-X” suggests basepairs are stacked across the branchpoint. Neither applies to this structure.

We appreciate this insightful comment. We have rephrased the illustration of the HJ as an open planar and X-shaped conformation with an overall two-fold symmetry.

11) Line 136: please correct the typo “Asp230”. Should be Asp253.

Point accepted. We have modified the typo.

12) Line 144-146: please rephrase. The position of the scissile phosphate cannot be known a priori since the branchpoint could take three distinct positions due to homology. Moreover, the CATG core is not cleaved by NtMOC1, and experiments in ref. 19 map the cleavage sites of a HJ containing a CCGG core. However its position can be inferred from its proximity to the enzyme’s active site in the structure.

Point accepted. In the revised manuscript, we have added another crystal structure of a cleavage inactive NtMOC1 in complex with HJ with a CCGG core. The complex structures of NtMOC1-HJ with CATG or CCGG cores could be well aligned. We have rephrased this section (Lines 116-127).

13) Line 156, 164: the contact with Lys225 appears to be through its main chain amine (fig. 4d), hence it should be anticipated that mutating this residue won’t affect activity.

Yes. Lys225 contacts the phosphate group of nucleotide C8 (Strand F) through the amine group of the main chain. Here, mutant of K225D could be used as a control that remains comparable HJ binding and cleavage activity as the WT NtMOC1 (Supplementary Figure 12).

14) Line 158: *“non-exchanging DNA strands”*. Should be *“exchanging DNA strands”*.

Point accepted. We have modified this demonstration.

15) Line 259-260: replace *“annealed in boiled water”* with *“incubated in boiled water”* and provide the composition of the annealing buffer.

Point accepted. We have modified the demonstration and provided the composition of the annealing buffer (25 mM Tris-Cl, pH 8.0, 150 mM NaCl) in the revised manuscript.

16) Line 268: *the crystallised protein is a quadruple mutant. How does its activity compare with that of the wild type enzyme?*

We appreciate this insightful suggestion. The quadruple mutant exhibit comparable binding and cleavage activities compared with the wild-type NtMOC1. We have added this result to supplementary Figure 7.

Supplementary Figure 7. HJ binding and cleavage activity of NtMOC1 mutants used for co-crystallization with HJs. a, HJ binding activity of NtMOC1 mutants. **b,** HJ cleavage activity of NtMOC1 mutants. WT, wild-type NtMOC1. tetraMut, NtMOC1 (I112V; Q162K; E235Q; 239Q). inactMut, NtMOC1 (I112V; Q162K; E235Q; 239Q; D116A; E175A; D253A; E258A). For the HJ binding and cleavage assay, X2 (CCGG) was used as the HJ substrate. The final concentration of HJ in each lane is 250 nM. Five gradients with increasing concentrations (0, 125, 250, 500, and 1000 nM) were applied for each protein sample. The reactions were resolved by native PAGE and visualized by GelRed staining.

17) Fig. 4: please reposition the yellow rectangles so that they actually encompass the area depicted in fig. 4b.

Point accepted. We have modified the yellow rectangles in Fig. 4b.

18) Sup. Fig. 2: please indicate the theoretical MW of a monomer.

Point accepted. We have indicated the theoretical MW of a monomeric NtMOC1 in the figure legend of Supplementary Figure 5b.

We truly thank this reviewer for his/her constructive comments.

REVIEWERS' COMMENTS:

Reviewer #1 (Remarks to the Author):

Junjie Yan, Sixing Hong, Zeyuan Guan, Wenjing He, Delin Zhang and Ping Yin resubmitted their paper entitled "Structural insights into sequence-dependent Holliday junction resolution by the chloroplast resolvase MOC1". The authors responded precisely to the concerns that were raised in my review on the previous version, and the revised paper is improved especially in the Discussion. I would like to recommend to accept this paper for publication.

Reviewer #2 (Remarks to the Author):

The authors have adequately addressed the comments from this reviewer. In particular, the electron density around the DNA binding/catalytic site provides strong support for the conclusions concerning the molecular interactions involved in recognition.

Reviewer #3 (Remarks to the Author):

The authors have addressed all of my questions in their revised manuscript.

They may wish to moderate their statement in lines 59, 152 and 237 that the structures present an "active" or "catalytic" configuration given that they contain either an active site mutant or a non-cognate DNA and no cleavage is observed.

There is a small typo on lines 82 and 84: "X2(CCGG)" should be "X2(CGCG)".

Response to referee's comments:

Reviewer #1

Junjie Yan, Sixing Hong, Zeyuan Guan, Wenjing He, Delin Zhang and Ping Yin resubmitted their paper entitled "Structural insights into sequence-dependent Holliday junction resolution by the chloroplast resolvase MOC1". The authors responded precisely to the concerns that were raised in my review on the previous version, and the revised paper is improved especially in the Discussion. I would like to recommend to accept this paper for publication.

This referee thought that we have precisely responded to his/her concerns. We thank this referee for his/her support.

Reviewer #2

The authors have adequately addressed the comments from this reviewer. In particular, the electron density around the DNA binding/catalytic site provides strong support for the conclusions concerning the molecular interactions involved in recognition.

This referee thought that we have adequately addressed his/her comments in the revised manuscript. We thank this referee for his/her support.

Reviewer #3

The authors have addressed all of my questions in their revised manuscript. (1) They may wish to moderate their statement in lines 59, 152 and 237 that the structures present an "active" or "catalytic" configuration given that they contain either an active site mutant or a non-cognate DNA and no cleavage is observed. (2) There is a small typo on lines 82 and 84: "X2(CCGG)" should be "X2(CGCG)".

This referee thought we have addressed all his/her questions in the revised manuscript. He/she also raised a minor comment and pointed out a small typo.

(1) Thanks for this insightful comment. We have removed these related demonstrations of "active" or "catalytic" in the revised manuscript.

(2) Point accepted. We have modified this typo in this revised manuscript.

We thank this referee for his/her support.